# NLRP3 lacking the leucine-rich repeat domain can be fully activated via the canonical inflammasome pathway

Iva Hafner-Bratkovič [1,2], Petra Sušjan [1], Duško Lainšček[1], Ana Tapia-Abellán[3], Kosta Cerović [1], Lucija Kadunc[1], Diego Angosto-Bazarra[3], Pablo Pelegrin [3] & Roman Jerala [1,2]

NLRP3 is a cytosolic sensor triggered by different pathogen- and self-derived signals that plays a central role in a variety of pathological conditions, including sterile inflammation. The leucine-rich repeat domain is present in several innate immune receptors, where it is frequently responsible for sensing danger signals and regulation of activation. Here we show by reconstitution of truncated and chimeric variants into $Nlrp3^{-/-}$ macrophages that the leucine-rich repeat domain is dispensable for activation and self-regulation of NLRP3 by several different triggers. The pyrin domain on the other hand is required to maintain NLRP3 in the inactive conformation. A fully responsive minimal NLRP3 truncation variant reconstitutes peritonitis in $Nlrp3^{-/-}$ mice. We demonstrate that in contrast to pathogen-activated NLRC4, the constitutively active NLRP3 molecule cannot engage wild-type NLRP3 molecules in a self-catalytic oligomerization. This lack of signal amplification is likely a protective mechanism to decrease sensitivity to endogenous triggers to impede autoinflammation.

[1] Department of Synthetic Biology and Immunology, National Institute of Chemistry, Hajdrihova 19, 1000 Ljubljana, Slovenia. [2] EN-FIST Centre of Excellence, Trg Osvobodilne fronte 13, 1000 Ljubljana, Slovenia. [3] Molecular Inflammation Group, Biomedical Research Institute of Murcia (IMIB-Arrixaca), University Clinical Hospital Virgen de la Arrixaca, Carretera Buenavista s/n, 30120 El Palmar, Murcia, Spain. Correspondence and requests for materials should be addressed to I.H.B. (email: iva.hafner@ki.si) or to R.J. (email: roman.jerala@ki.si)

Nucleotide-binding domain (NBD) and leucine-rich repeat (LRR)-containing proteins (NLRs) are cytosolic sensors of pathogen-associated molecular patterns (PAMPs) and danger-associated molecular patterns (DAMPs). Activation of some NLRs leads to the assembly of an inflammasome, comprising a sensor component such as NLRP3 (NLR family protein containing a pyrin domain 3), an adaptor ASC (apoptosis-associated speck-like protein containing a caspase activation and recruitment domain, CARD), and the effector pro-caspase-1. Inflammasome formation results in autoproteolytic caspase-1 activation. Caspase-1 further cleaves various substrates, including cytokines pro-IL-1β and pro-IL-18 and gasdermin D, which induces pyroptotic cell death by pore formation[1,2].

The physiological role of the NLRP3 inflammasome is enigmatic[3], as it is triggered by a wide range of chemically and morphologically diverse activators, from small molecules (such as ATP and nigericin[4,5]) to particulate triggers silica[6,7], amyloid β[8], or prion protein fibrils[9] and others. NLRP3 inflammasome has been linked to diabetes[10] and Alzheimer's disease[11,12] and missense substitutions in the NLRP3 protein-encoding gene result in cryopyrin-associated periodic syndromes (CAPS)[13]. Although the contribution of NLRP3 to various pathologies has been described, little is known about the molecular mechanism of NLRP3 inflammasome assembly. K$^+$ efflux is likely the physiological process downstream of diverse activators[14], apart from imiquimod[15]. How K$^+$ efflux induces assembly of the NLRP3 inflammasome remains elusive.

NLR proteins usually contain an interaction domain (pyrin, CARD, BIR) at the N-terminus and LRR domain at the C-terminus. The central NBD with NBD-associated domains (helical domain 1, HD1; winged helical domain, WHD, and helical domain 2, HD2) is crucial for oligomerization of NLRs upon activation. The central domain is also known as the NACHT domain (which comes from present in NAIP, CIITA, HET-E, and TP-1) and comprises NBD and the first two associated domains[16]. Molecular structures of homologous proteins NLRC4[17] and NLRC2[18] in an inactive conformation were determined, advancing the structural knowledge of NLRs. The structure of mouse NLRC4 lacking the N-terminal CARD domain reveals that autoinhibition is provided by extensive intramolecular interactions between different domains and interactions with ADP[17] (reviewed in[19]). There are substantial structural differences between NLRC4 and NLRC2, mostly in the HD2 domain and its orientation toward the NBD-HD1-WHD[18]. The structure of NLRC4 in an inactive form was used to propose that the LRR domain needs to separate from the central NBD-HD1-WHD module in order for oligomerization to take place[17]. Later studies proposed the model of NLRC4 oligomerization based on cryoelectron microscopy studies of the disk-like structures of NAIP2-NLRC4 (NAIP5-NLRC4) oligomers[20–22]. NLRC4-containing wheel-like particles are substoichiometric, revealing 11- or 12-fold symmetry[20,21]. Interestingly, NLRC4 is not the receptor for bacterial ligands, which are sensed by the members of the NAIP family through their HD1-WHD-HD2 region[23]. Additionally, flagellin makes contacts with the BIR1 domain and its N-terminus, as well as with a small part of LRR of NAIP5[24]. The binding of bacterial ligand to the respective member of the NAIP family causes the recruitment of the NLRC4 molecule and the subsequent release of autoinhibitory interactions[20,21]. This complex recruits additional NLRC4 molecules in prion-like polymerization leading to the assembly of substoichiometric oligomers, which enables amplification of the signal and highly sensitive immune surveillance. With the exception of the structure of the PYD domain[25,26], information about the structure of NLRP3 in an inactive or active conformation, as well as information about whether NLRP3 is also capable of similar self-amplification, is lacking.

The differences in the available homologous structures and proposed mechanisms of apoptosome and inflammasome assembly[17,18,27–29], along with the unknown identity of a direct NLRP3 ligand, make it difficult to propose a reliable functional model of NLRP3 activation. Thus, we decided to perform extensive mutagenesis of NLRP3 to determine the role of particular domains of NLRP3 in sensing triggers and in the NLRP3 inflammasome assembly.

We show that the LRR domain is not autoinhibitory and is clearly not an activator-sensing domain given that variants lacking the LRR domain are not constitutively active and respond to triggers to a similar degree as full-length NLRP3. A minimal fully active NLRP3 variant is defined, and cells harboring this variant reconstitute peritonitis in Nlrp3$^{-/-}$ mice, demonstrating the full proinflammatory functionality of minimal NLRP3. Further we show that the replacement of the pyrin domain with the CARD domain of ASC or NLRC4 led to constitutive activation, suggesting that the pyrin domain likely interacts with other domains of NLRP3 and thus, contributes to autoinhibition. Finally, we demonstrate that pathological constitutively active NLRP3 mutants are unable to engage the wild-type NLRP3 variant in a functional inflammasome complex in physiological conditions, strongly arguing against the prion-like polymerization mechanism of NLRP3 activation.

## Results

**LRRs do not stabilize the inactive conformation of NLRP3.** We decided to first define the contribution of the LRR domain in autoinhibition and activation of NLRP3. To design truncation variants of NLRP3 that map domain boundaries, bioinformatic tools (including homology modeling based on the available structures of NLRC4 and NLRC2, LRR prediction tools, and previous publications) were taken into account[16–18,30–33], (Supplementary Fig. 1 a–d). To estimate the role of the NLRP3 LRR domain and specific LRRs, 11 truncation variants of the mouse NLRP3 sequence were designed within the predicted LRR region (Fig. 1a). The 731 (1–731) variant of mouse NLRP3 lacks all the predicted LRRs, while the nine other truncations (1–766 to 1–996) each contain one predicted LRR more than the shorter one. Overexpression of NLRP3 inflammasome components often leads to constitutive activation in the absence of the trigger, particularly due to the propensity of ASC for aggregation[34,35]. To eliminate the physiologically irrelevant overexpression artifacts, constructs encoding LRR truncated variants were integrated into the genome of NLRP3$^{-/-}$ immortalized BMDMs using retroviral vectors, where NLRP3 variant gene is under the doxycycline-inducible promoter to adjust the expression level (Supplementary Fig. 2a–e). This design appropriately reproduces NLRP3 inflammasome activation; wild-type NLRP3 is activated only in the presence of inflammasome trigger nigericin, while CAPS-associated NLRP3 variant R258W is constitutively active (Supplementary Fig. 2c)[36].

Introduction of NLRP3 variant did not result in the substantial changes in the release of inflammasome-independent cytokine (IL-6 and TNF-α) (Supplementary Fig. 3 a, b). Truncation of one to six LRRs of NLRP3 from the C-terminus led to a loss of the response to NLRP3 canonical soluble triggers, such as nigericin (Fig. 1b), and to the particulate triggers silica and alum (Fig. 1c). This effect, however, could be at least partially attributed to the lower protein levels of the truncation variants 1–850 to 1–996 (Fig. 1d). We show that two truncated variants (1–879 and 1–965) failed to respond to nigericin even at the highest allowed doxycycline concentration (Supplementary Fig. 4a), yet the protein levels of those variants were still below the protein level of full-length NLRP3 needed for the proper response to nigericin

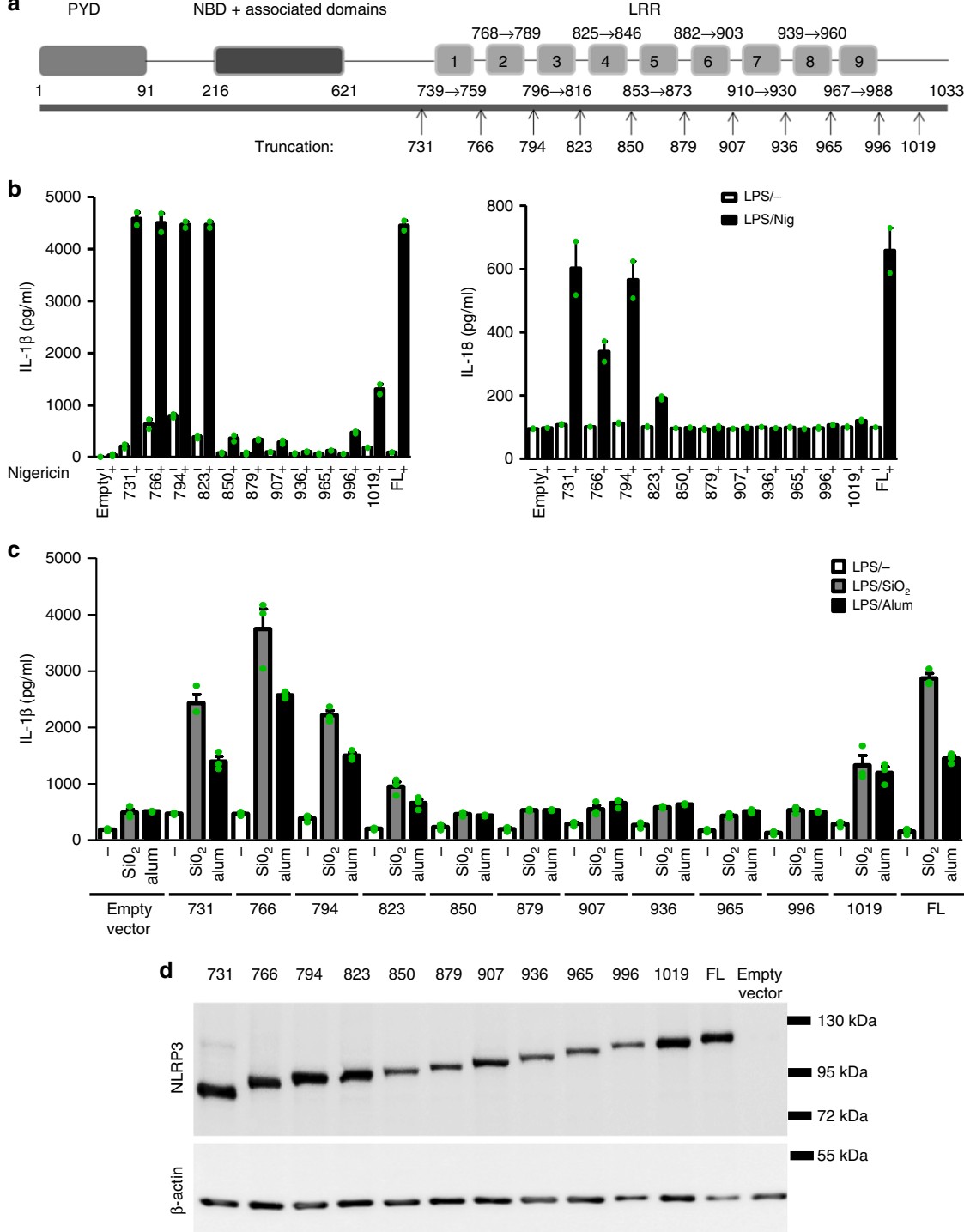

**Fig. 1** The LRR domain is not important for activation of NLRP3. **a** Schematic representation of truncation variants based on the predicted domain organization of the NLRP3. Variants are labeled either as 1-last amino acid residue or by just the last amino acid residue number. **b** The response of truncated variants with variable numbers of LRRs to nigericin (IL-1β, left; IL-18, right). Immortalized mouse bone marrow–derived macrophages from *Nlrp3*$^{-/-}$ mice with stable integration of the designated constructs were primed with LPS (100 ng/ml) and doxycycline (0.05 μg/ml) for 11 h. After priming, the medium was exchanged, and the cells were stimulated with nigericin (5 μM) for 1 h. Representative of 3 independent experiments is shown. The mean and the s.e.m. are shown of 2 biological replicates which are combined to show expression in **d**. **c** Cells were primed as in **b**. After priming, SiO$_2$ (0.2 mg/ml) and alum (0.5 mg/ml) were added for 6 h. Representative of 3 independent experiments is shown. The mean and the s.e.m. of 3 biological replicates are shown. **d** Western blot analysis of primed and doxycycline-treated iBMDMs. NLRP3 was detected with the Cryo-2 antibody, which binds to the pyrin domain. Representative of 3 independent western blots is shown

(Supplementary Fig. 4b–d). While the protein level of full-length NLRP3 was stable after 6 h, the protein levels of 1–879 and 1–965 were decreased upon the removal of doxycycline (as judged by the difference in the expression level at 3 and 6 h post doxycycline removal) and further decreased by inhibition of protein synthesis (Supplementary Fig. 5a). The levels of 1–879 and 1–965 were partially rescued by the use of proteasome inhibitors (Supplementary Fig. 5b). Results, thus, suggest that truncation of the C-terminal LRRs leads to clearance of NLRP3 variants by proteasome degradation and instability of these NLRP3 variants could at least partially contribute to their unresponsiveness.

The four shortest NLRP3 truncations (731–823), containing 0, 1, 2, or 3 predicted N-terminal LRRs, however, maintained full responsiveness to canonical NLRP3 activators (Fig. 1b, c) and exhibited an expression level similar to that of full-length NLRP3 (Fig. 1d). Therefore, NLRP3 lacking all predicted LRRs responded to nigericin, alum, and silica particles similarly to full-length NLRP3. The basal release of IL-1β in macrophages bearing truncated NLRP3 variants was low and comparable to those expressing the full-length NLRP3 protein (Fig. 1c). Therefore, these results show that truncated variants of NLRP3 with an uncompromised expression level respond to canonical NLRP3 triggers to a comparable degree as the full-length protein, suggesting that the LRR domain is not necessary for sensing or for assembling a functional NLRP3 inflammasome, and that the LRR domain does not contribute significantly to stabilization of the NLRP3 inactive state.

**The minimal responsive NLRP3 variant.** A previous study using mutagenesis of members of the NAIP family located the regions responsible for binding of bacterial ligands N-terminally to the LRR domain[23]. We decided to perform eight further truncations of NLRP3 in the region between the LRR domain and the NBD-associated domains (1–620 to 1–720) to define the role of this region in activation of the NLRP3 inflammasome (Fig. 2a). Those truncations were designed with the help of an NLRP3 three-dimensional model based on NLRC4 (Supplementary Fig. 1c) to take into account the predicted secondary structural elements and motifs, such as charged clusters. Two additional NLRP3 truncated variants were prepared that consist of the PYD-NBD-HD1-WHD (1–541, Fig. 2a, and 1–572, Supplementary Fig. 6a). Truncation of the LRR domain leads to partially constitutive activation of NLRP3 in HEK293[37] or THP-1 cells[38]. This was not the case for the NLRP3 truncations used in the present study, where the level of expression was comparable to the physiological levels of endogenous NLRP3 upon LPS priming (Supplementary Fig. 6b). NLRP3 truncated variants ending in amino acid residues 686, 695, 710, and 720 retained the full ability to release IL-1β (Fig. 2b), while shorter NLRP3 variants (Fig. 2b, Supplementary Fig. 6a) failed to induce the release of IL-1β in response to nigericin, despite sufficient expression levels (Fig. 2c, Supplementary Fig. 6a, b). The selected variants, particularly the minimal responsive NLRP3 (1–686), hereafter called miniNLRP3 and the longest non-responsive (1–665) variant were tested for caspase-1 activation in response to nigericin. MiniNLRP3 and full-length NLRP3 activated caspase-1 after treatment with nigericin (Fig. 2d) and responded to the particulate activators silica and alum with release of IL-1β (Fig. 2e), while the 1–665 variant failed to activate caspase-1 and release IL-1β, regardless of the trigger (Fig. 2d, e). Some small molecule activators of the NLRP3 inflammasome, such as imiquimod, were reported to activate NLRP3 independently of K$^+$ efflux[15]. We found that miniNLRP3 is also activated in response to imiquimod (Fig. 2f). Introduction of NLRP3, either wild-type or truncation variant did not result in large changes in the release of inflammasome-

independent cytokine (IL-6, TNF-α) levels (Supplementary Fig. 7 a–c). The introduction of CAPS-associated mutations R258W and T348M induced constitutive maturation of IL-1β, while the amount of basal (LPS-treated in the absence of an NLRP3 trigger) IL-1β released by the truncation variants was much lower and comparable to the basal release from the empty vector transduced cells (Fig. 2b, e, f).

To exclude the possibility that truncated variants and full-length NLRP3 have different kinetic behavior, activation by nigericin was monitored for different durations of the priming step. MiniNLRP3 exhibited the same time-dependence as full-length NLRP3, while the 1–665 NLRP3 truncation remained defective under all tested conditions (Supplementary Fig. 8a). A shorter LPS priming duration (Supplementary Fig. 8b) and a different priming agent (Pam2CSK4) (Supplementary Fig. 8c) still provided sufficient priming for the miniNLRP3 to respond to nigericin and ATP by IL-1β release. We also demonstrated that miniNLRP3 has the same life-time and stability as full-length NLRP3 (Supplementary Fig. 8d, e) and that nigericin activation of miniNLRP3 and full-length NLRP3 was abrogated by the inhibition of ROS and K$^+$ efflux (Fig. 2g).

Activation of the NLRP3 inflammasome is regulated by posttranslational modifications such as ubiquitination and phosphorylation. Therefore, the role of the LRR might be as the site of posttranscriptional regulation. The majority of posttranslationally modified sites, for example, by protein phosphatase 2 A (S3)[39], Jun N-terminal kinase (S194)[40], protein kinase A[41,42], and protein kinase D (S293)[43], are present in miniNLRP3. Some of the regulatory posttranslational modifications target the LRR domain and sites not present in miniNLRP3. For example, FBXL2 targets NLRP3 at K687 for ubiquitination and degradation[44]. The K687 site is missing in miniNLRP3; however, the observed cytosolic protein levels of miniNLRP3 and NLRP3(1–695) were, nevertheless, similar. Dephosphorylation of Y859p by protein tyrosine phosphatase non-receptor 22 (PTPN22)[45] and BRCC3 deubiquitination of LRR domain[46] are also missing in miniNLRP3. Specific inhibitors of mouse PTPN22 are not available hampering simple experiments to probe the effect of PTPN22 on miniNLRP3. In contrast, the deubiquitinase inhibitor G5 concentration dependently inhibited both miniNLRP3 and the full-length NLRP3 activation (Fig. 2h), which is not surprising as the NACHT region is also deubiquinated by BRCC3[46]. Although the life-time of the two proteins in the resting state was similar we cannot completely exclude differential regulation (e.g., through Y861p dephosphorylation by PTPN22[45]) of full-length NLRP3 compared to the truncated miniNLRP3.

Several studies have identified NEK7 kinase binding as crucial for activation of the NLRP3 inflammasome[15,35,47,48]. Although the NEK7 binding site has been previously mapped to the C-terminus of the LRR domain[35], NEK7 was pulled down by 1–665 and miniNLRP3 that lack the predicted LRRs (Supplementary Fig. 9a). In LPS-primed macrophages, after stimulation with nigericin, NEK7 pulldown was increased only in the case of inflammasome-sufficient variants (Supplementary Fig. 9b). The present results suggest that at least one other NEK7 binding site is located within the N-terminal domains of NLRP3. This finding is corroborated by a recent study showing that oridonin inhibits activation of NLRP3 by disrupting NEK7 binding through covalent modification of the NLRP3 at Cys279[49].

In recent years, several compounds were shown to dampen inflammation by inhibiting activation of inflammasomes. Here, we show that activation of miniNLRP3 and full-length NLRP3 is inhibited to a similar degree by the NLRP3-specific small molecule inhibitors MCC950[50] and CY-09[51] and the non-specific inflammasome inhibitor shikonin[52] (Fig. 2h). In summary, NLRP3 (1–686) is a minimal responsive variant able to

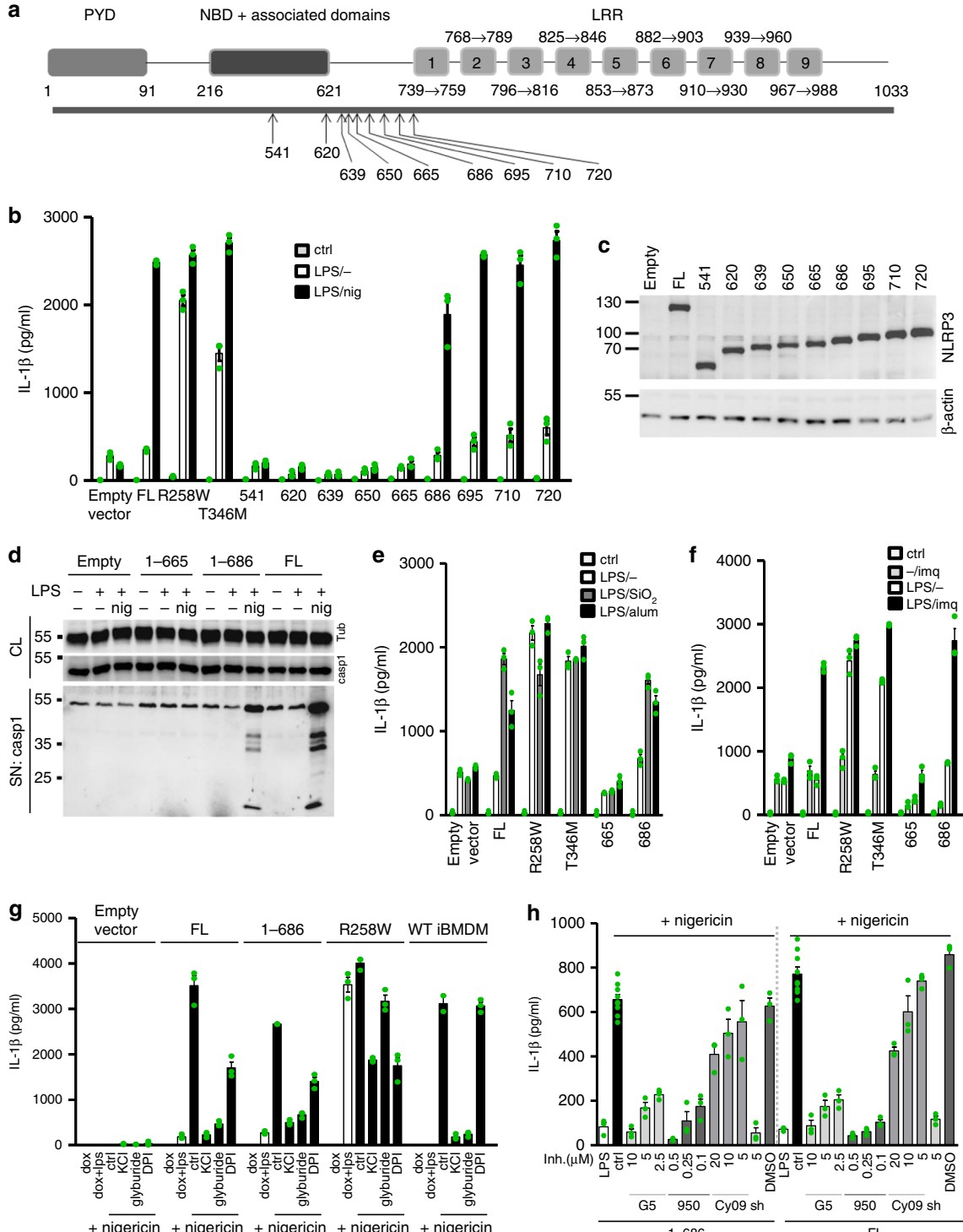

**Fig. 2** Determination of the minimal fully responsive NLRP3 variant. **a** Schematic representation of truncation variants based on the predicted domain organization of the NLRP3. **b** Immortalized mouse bone marrow–derived macrophages from NLRP3-deficient mice with stable integration of the designated constructs were primed with LPS (100 ng/ml) and doxycycline (1 µg/ml) for 11 h. After priming, the medium was exchanged, and the cells were stimulated with nigericin (5 µM) for 1 h. **c** Western blot of NLRP3 variants. **d** Selected cell lines were stimulated as in **b**, and caspase-1 autoproteolytic activation was followed by western blotting. **e** Selected cell lines were primed as in **b**. After priming, SiO$_2$ (0.2 mg/ml) and alum (0.5 mg/ml) were added for 6 h. **f** Cell lines were primed as in **b**. After priming, the medium was replaced, and imiquimod (20 µg/ml) was added for 24 h. **g** Stable $Nlrp3^{-/-}$ iBMDMs harboring different NLRP3 variants and wild-type iBMDMs were primed with LPS and doxycycline for 11 h, after which the medium was replaced, and the inhibitors KCl (130 mM), glyburide (100 µM), and DPI (50 µM) were added for 0.5 h before nigericin was added (5 µM). **h** $Nlrp3^{-/-}$ iBMDMs were primed as before, inhibitors G5, MCC950 (950), Cy-09 and shikonin (sh) were added prior nigericin stimulation. Representative of 3 (**b**, **e**, **f**) or 2 (**c**, **d**, **g**, **h**) independent experiments is shown. The mean and the s.e.m. of 3 (**b**, **e**–**h**) or 8 (ctrl in **h**) biological replicates are shown. 1–686 variant corresponds to MiniNLRP3

respond to canonical NLRP3 activators with similar kinetics and under similar regulation as the full-length NLRP3 protein.

**Activation of human miniNLRP3 leads to ASC speck formation.** Human and mouse NLRP3 proteins are highly conserved (83% identity), and human NLRP3 was previously shown to reconstitute the inflammasome response in *Nlrp3*[−/−] mouse macrophages[53]. To determine whether human variants have the same properties as those originating from mouse NLRP3, human NLRP3 truncations 1–667 and 1–688, corresponding to the mouse non-active variant 1–665 and miniNLRP3, were generated and stably integrated into mouse *Nlrp3*[−/−] iBMDMs. As in the case of mouse variants, human 1–688 variant responded to nigericin and ATP comparably to full-length human NLRP3, while the truncation 1–667 was nonresponsive in terms of the release of IL-1β (Fig. 3a, left) and inducing macrophage pyroptosis (Fig. 3a, right). Comparable levels of pro-caspase-1, pro-IL-1β, and NLRP3 variants were observed in cell lysates upon LPS priming (Fig. 3b), but only miniNLRP3 and full-length human NLRP3 responded to nigericin with the maturation and release of caspase-1 and IL-1β (Fig. 3b). The secretion of TNF-α and IL-6 induced by LPS was comparable in all cell lines (Fig. 3c). We were further interested in the signaling events upstream of activation of caspase-1, such as the formation of ASC specks, and found that the 1–667 NLRP3 truncation failed to induce formation of ASC specks upon treatment with nigericin (Fig. 3d). Activation of miniNLRP3 resulted in a similar formation of ASC specks as for full-length NLRP3 (Fig. 3e) and as observed upon stimulation of immortalized macrophages from wild-type mice (Supplementary Fig. 10), showing that miniNLRP3 is able to form a proper NLRP3 inflammasome, leading to the formation of ASC specks. These results demonstrate that human miniNLRP3, similar to mouse miniNLRP3, is not constitutively active in the absence of the predicted LRR domain but is fully responsive to nigericin with comparable formation of ASC specks, activation of caspase-1, release of IL-1β, and induction of pyroptosis as full-length NLRP3.

To facilitate NLRP3 visualization 1–665, miniNLRP3, and full-length mouse NLRP3 were tagged with YFP at the C-terminus. The functionality of the tagged versions was comparable to that of the untagged versions (Fig. 4a). The NLRP3 1–665 variant did not support activation of the inflammasome (Fig. 4a) despite the fact that localization (Fig. 4b) and the level of expression (Fig. 4b, c) of the YFP-tagged NLRP3 version in the resting macrophages were similar to those of miniNLRP3 and full-length NLRP3. Upon stimulation with nigericin, the 1–665 NLRP3 truncation remained dispersed through the cytosol, while miniNLRP3 and full-length NLRP3 clustered into a perinuclear region (Fig. 4b). Additionally, miniNLRP3 and full-length NLRP3, but not 1–665, bound ASC after stimulation with nigericin (Fig. 4c).

**MiniNLRP3 facilitates peritonitis in *Nlrp3*[−/−] mice.** At this point, the results clearly indicated that miniNLRP3 truncation is fully functional in cell cultures, while a 21 amino acid shorter variant is incompetent in inflammasome activation. We were interested in whether cells harboring the selected NLRP3 truncations are functional in vivo and are able to induce peritonitis in *Nlrp3*[−/−] mice. Injection of LPS-primed immortalized macrophages carrying an empty vector, 1–665 inactive NLRP3 truncation, miniNLRP3, or full-length NLRP3 resulted in increased IL-1β (Supplementary Fig. 11) and neutrophil recruitment in *Nlrp3*[−/−] mice for miniNLRP3 and full-length NLRP3 (Fig. 4d), demonstrating that miniNLRP3 acts similarly to full-length NLRP3. While establishing this mouse peritonitis model, we observed that detachment of primed iBMDMs was sufficient to

induce activation of the inflammasomes in cells expressing full-length NLRP3 and in cells with miniNLRP3, which is not surprising as activation of NLRP3 inflammasome has been previously linked to changes in cell volume[54] and activation of ion channels (reviewed in[55]).

**Inactive NLRP3 truncation 1–665 senses triggers.** The segment between 665 and 686 contains several amino acid residues that could potentially contribute to posttranslational regulation and activation of the NLRP3 inflammasome (Fig. 5a). Several studies emphasized the effect of ROS on inflammasome activation[56,57]. The region from 665 to 686 contains two cysteine residues, Cys667 and Cys671, which were mutated to Ala. We also decided to mutate Cys6 and Cys104, as those residues formed a disulfide bridge in the crystal structure of the pyrin domain[26], which might be involved in a potential role of NLRP3 in redox sensing. In addition, the positively charged residues His672 and Arg673 were mutated to Ala in order to analyze the potential role of charged interactions. However, none of the single or double point mutations changed the release of IL-1β in response to nigericin (Fig. 5b), suggesting that neither cysteines nor a positively charged cluster in the 665 to 686 segment play a role in NLRP3 inflammasome activation or in stabilizing interactions as no constitutive activation was observed. NLRP3 requires ATP binding for signaling[58]. Therefore, it is expected that miniNLRP3 should retain the ATP-binding propensity. However, further truncation, which yields inactive 1–665, might cause the formation of a protein that is not folded correctly. NLRP3-specific inhibitor CY-09 competes with ATP for binding to NLRP3[51]. We found that 1–665, miniNLRP3, and full-length NLRP3 bind to ATP-agarose beads and that this binding was decreased in the presence of CY-09 (Fig. 5c). This result demonstrates that despite the inability of 1–665 to induce the functional inflammasome, the nucleotide-binding module of 1–665 is intact.

To elucidate the conformational rearrangements of 1–667, 1–688, and full-length human NLRP3 in live cells, we employed the bioluminescence resonance energy transfer (BRET) technique[54,59,60]. BRET is observed when donor and acceptor molecules are within a distance < 10 nm. First, we decided to follow changes in the distances within the same molecule; thus, the donor (luciferase) and the acceptor (YFP fluorescent protein) were positioned on the C- and N-termini of NLRP3, respectively, as described previously[61]. We determined that luciferase/YFP-tagged miniNLRP3 and full-length NLRP3 variants expressed in NLRP3[−/−] macrophages induced release of IL-1β upon activation by nigericin (Supplementary Fig. 12a, b[60]). The BRET signal observed for full-length NLRP3 in the resting state was higher than the BRET signal for miniNLRP3 (Fig. 5d), but both were intramolecular, as deduced from the titration experiments (Supplementary Fig. 12c–e). According to the molecular models, the LRR domain (Supplementary Fig. 1c, d) brings the donor and acceptor tags closer, which explains the lower BRET signal for miniNLRP3. The BRET signal for the 1–667 NLRP3 truncation was much lower than that for miniNLRP3 (Fig. 5d). Surprisingly, following the intramolecular BRET signal upon nigericin addition, we observed that even NLRP3 (1–667) responds to nigericin with a characteristic drop of the BRET signal, observed in both cell systems used (Fig. 5d, Supplementary Fig. 12f), with comparable slopes of the normalized signal drop for different NLRP3 variants (Fig. 5d, right).

These experiments demonstrated that 1–665 (human 1–667) is able to sense the physiological response triggered by nigericin and retains the propensity to bind to ATP, yet it fails to support inflammasome formation.

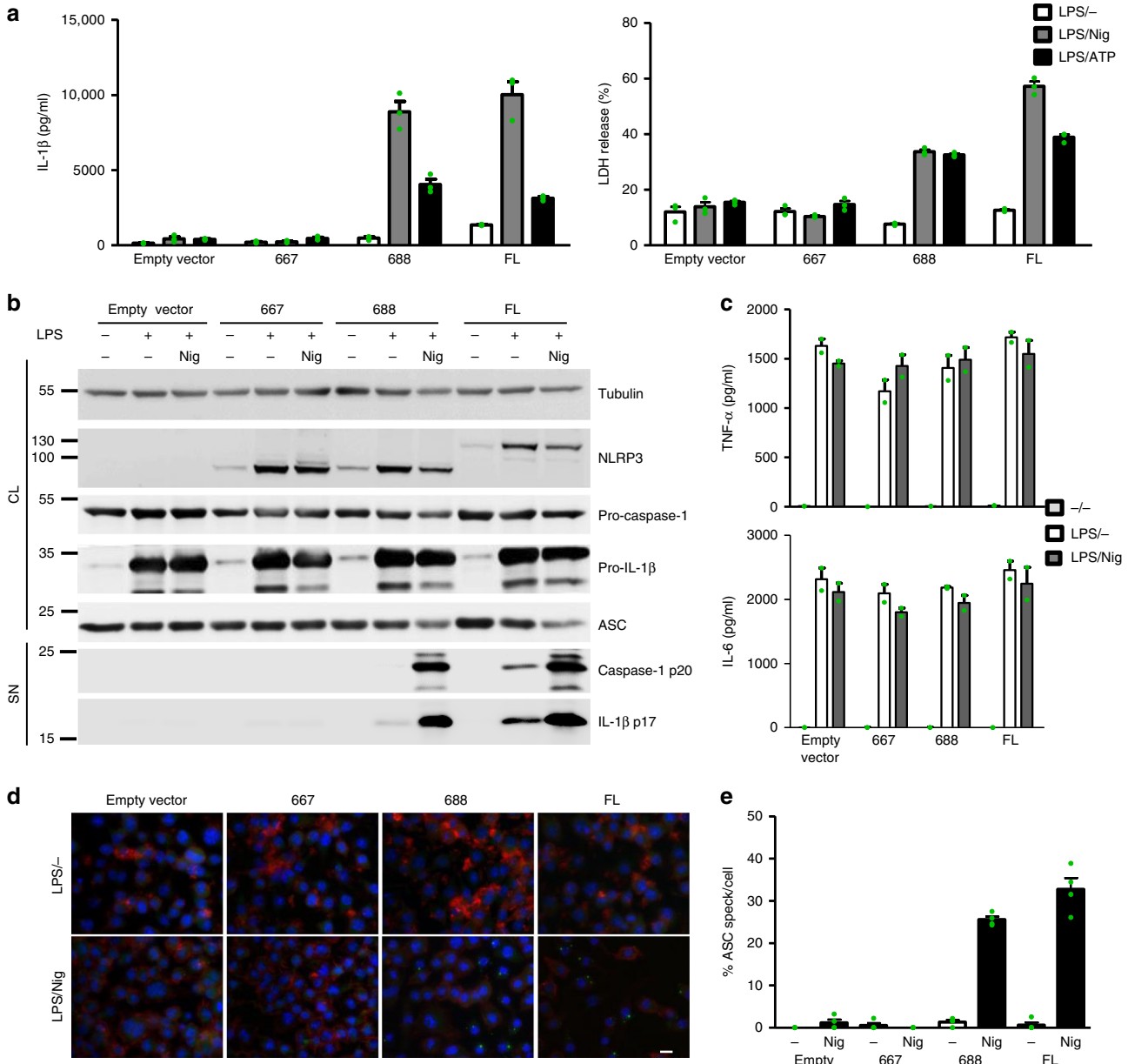

**Fig. 3** Human variants phenocopy corresponding mouse variants. **a** *Nlrp3*[−/−] iBMDMs with stably integrated human NLRP3 variants were primed with LPS (100 ng/ml) and doxycycline (1 μg/ml) for 11 h and stimulated with nigericin (10 μM) or ATP (5 mM) for 45 min. Supernatants were analyzed for IL-1β maturation (left) and LDH activity (right). **b** Cells were unprimed (doxycycline-treated) or primed (LPS, doxycycline) for 12 h and stimulated for 1 h, and cell supernatants were analyzed for mature IL-1β and caspase-1 p20 subunit. Cell lysates were analyzed for expression of pro-caspase-1, pro-IL-1β, ASC, and NLRP3 variant. **c** Supernatants from **b** were analyzed for TNF-α and IL-6 concentrations. **d**, **e** LPS-primed and nigericin-treated *Nlrp3*[−/−] iBMDMs with human NLRP3 were analyzed for the formation of ASC specks. Nuclei are depicted in blue (DAPI), ASC in green, and actin in red; the bar represents 10 μm. To provide an estimate of ASC speck formation (in %) (**e**), four random 138 × 110 μm² frames were recorded for each condition, the number of ASC specks was divided by the number of nuclei within each frame. Representative of 3 (**a**–**e**) independent experiments is shown. The mean and the s.e.m. of 3 (**a**) or 2 (**c**) biological replicates and 4 random frames (**e**) are shown. 1–688 variant corresponds to MiniNLRP3

**MiniNLRP3 supports CAPS-associated constitutive activity**. A possible explanation for the inability of 1–665 to support inflammasome assembly is the potential loss of the binding site for the putative direct ligand which acts downstream of canonical inflammasome triggers. As pathological variants are constitutively active, truncations were prepared on different pathologic mutant NLRP3 backgrounds. The 1–665 NLRP3 truncation was not constitutively active on the R258W (Fig. 6a) or T346M (Fig. 6b) background, while miniNLRP3 and full-length NLRP3 were constitutively active. This result demonstrates that miniNLRP3 is fully sufficient for inflammasome assembly and activation, while

1–665 and the shorter truncation of NLRP3 (1–572) lack the ability to assemble a functional inflammasome (Fig. 6b). The residues located C-terminally to the position 665 are not involved in trigger sensing but in the assembly of the functional inflammasome.

To monitor oligomerization or rearrangement of NLRP3 via intermolecular BRET, pairs of the acceptor (YFP) and donor (*Renilla* luciferase)-tagged NLRP3 variants were transfected into the HEK293 cell line and the BRET signal was followed in the resting state or after treatment with nigericin or buffer (Supplementary Fig. 13a–i). As reported previously, the resting

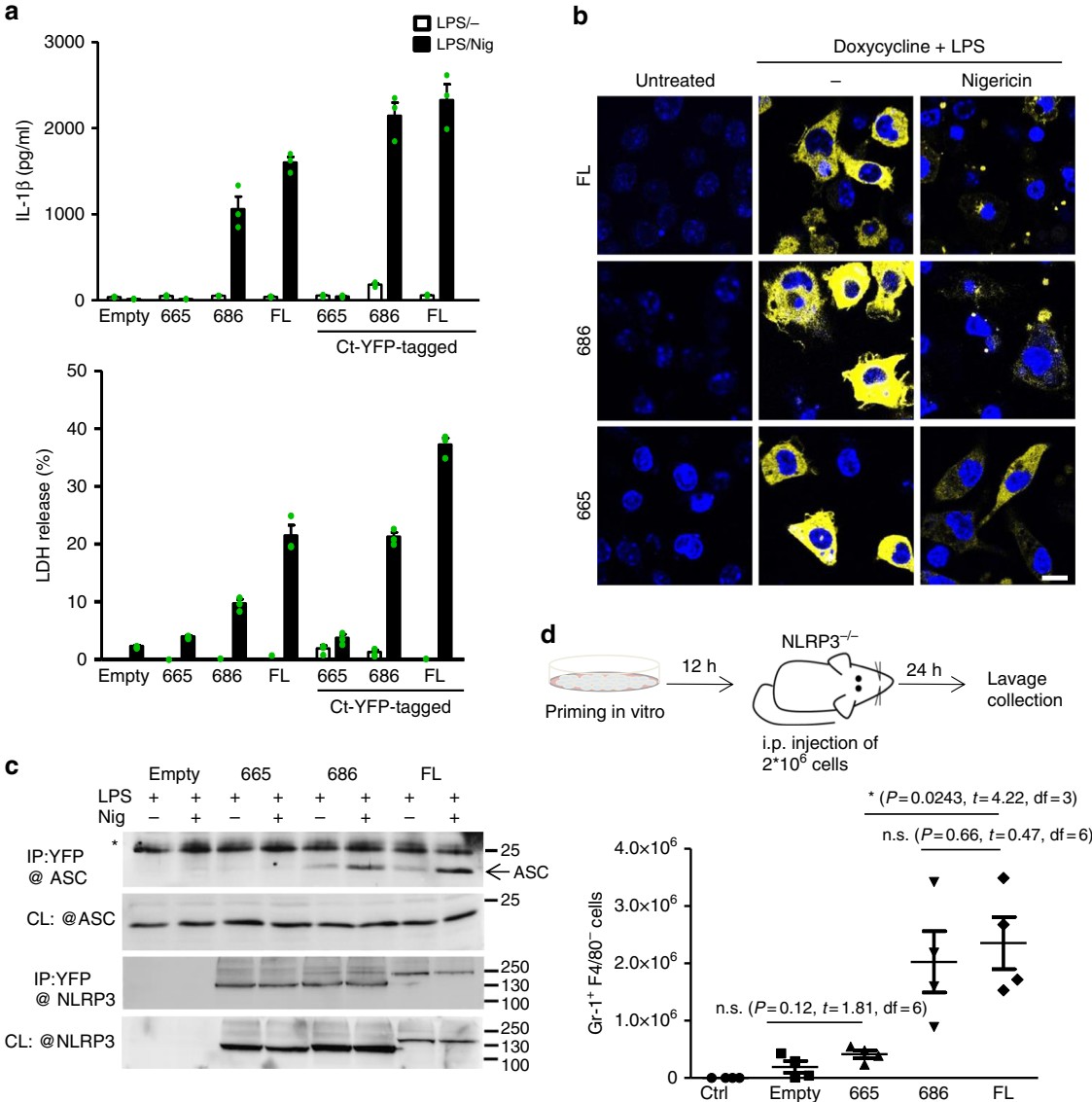

**Fig. 4** MiniNLRP3 clusters after stimulation and facilitates peritonitis in *Nlrp3*−/− mice. **a** Tagged and untagged versions of NLRP3 variants were primed for 11 h with LPS and doxycycline (0.5 μg/ml) and stimulated for 1 h with nigericin (5 μM). Supernatants were analyzed for IL-1β maturation and LDH activity. **b** Expression of murine NLRP3-YFP-tagged variants in dormant and activated conditions. Cells were untreated (ctrl), LPS and doxycycline-treated, and LPS, doxycycline and nigericin (5 μM)-treated (1 h). Nuclei are depicted in blue (Hoechst) and NLRP3-YFP variants in yellow; the scale bar represents 10 μm. **c** Cells were primed as in **a**. After priming, the pan-caspase inhibitor (Z-VAD-FMK) was added for 30 min before stimulation with nigericin for 30 min. Cells were lysed, and proteins were immunoprecipitated with antibodies against fluorescent protein, * non-specific band (likely light chain). Representative of 3 (**a**, **b**) or 2 (**c**) independent experiments is shown. The mean and s.e.m. of 3 biological replicates are shown **a**. **d** Primed and doxycycline-stimulated *Nlrp3*−/− iBMDMs carrying either the empty vector, 1–665, 1–686, or full-length mouse NLRP3 were injected into the peritoneal cavity of the *Nlrp3*−/− mice, and neutrophil infiltration was analyzed 24 h later. Four animals were used per condition. The mean and the s.e.m. are depicted. Statistical values: n.s. *p* > 0.05, **p* < 0.05. The two-tailed nonparametric *t*-test without/with Welch correction was used for pairwise comparison. 1–686 variant corresponds to MiniNLRP3

BRET signal of N-terminally YFP-tagged full-length NLRP3 was higher than of C-terminally YFP-tagged NLRP3 when BRET was followed in combination with C-terminally rLUC-tagged NLRP3 (Supplementary Fig. 13c)[54]. An increase in the BRET signal in the steady state after treatment with nigericin was observed only for the combinations of full-length NLRP3 and miniNLRP3 (YFP-FL, YFP-688, FL-YFP, and 688-YFP when paired with FL-rLUC (Supplementary Fig. 13c); YFP-688 and 688-YFP when paired with 688-rLUC, Supplementary Fig. 13d) indicating oligomer formation and conformational rearrangement upon nigericin treatment. These results show that full-length NLRP3 and miniNLRP3 can engage in the same complex, while 667 in any

orientation failed to form oligomers that would lead to a substantial BRET signal (Supplementary Fig. 13c–e). The results suggest that 1–667 fails to form the oligomer that is required for efficient recruitment and docking of ASC.

**PYD domain locks NLRP3 in inactive conformation**. The structure of NLRC4 without the CARD domain revealed that the inactive conformation of NLRC4 is stabilized by the adenosine diphosphate, which interacts with residues located in NBD and WHD[17,19]. Additionally, the HD-2 and LRR domains interact with the NBD domain, further locking NLRC4 into the inactive

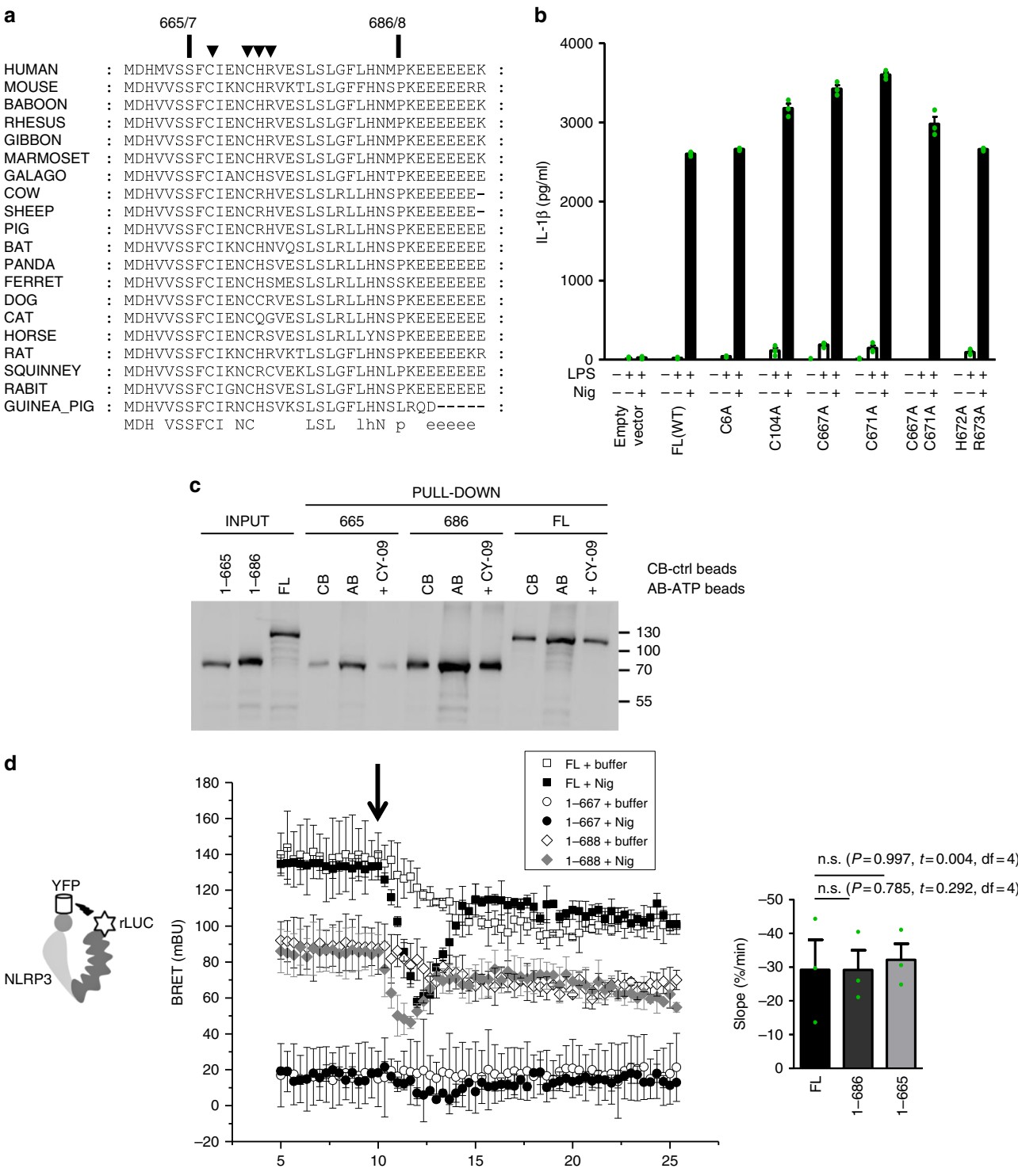

**Fig. 5** NLRP3 (1–667) senses nigericin-induced perturbation. **a** Alignment of NLRP3 protein sequences in the region between 1–665 and 1–686 from different species. **b** Single and double point mutants were transduced into $Nlrp3^{-/-}$ iBMDMs, and after selection, the cells were primed for 12 h and stimulated with nigericin (5 μM) for 1 h. **c** NLRP3 variants were expressed in HEK293. Cell lysates were incubated with control beads (CB) or ATP beads (AB). NLRP3 binding to ATP beads was also followed in the presence of CY-09 ( + CY-09) and detected by western blot. **d** Intramolecular BRET of the human NLRP3 variants was followed in the HEK293 cells after nigericin was added (left). The arrow indicates the time of injection of nigericin or buffer. The bar graph (right) represents the slope of the curve after nigericin addition. To calculate the slope, the signals were normalized to the resting BRET signal (the signal before nigericin addition) to annihilate differences in BRET responses. n.s.: $p > 0.05$. (**c**) Representative of 3 (**d**) or 2 (**b**, **c**) independent experiments is shown. The mean and the s.e.m. (**b**) or s.d. (**d**) of 3 biological replicates are shown. Individual points of **d** are depicted in Supplementary Fig. 24. MiniNLRP3 is represented by mouse 1–686 or human 1–688 variants

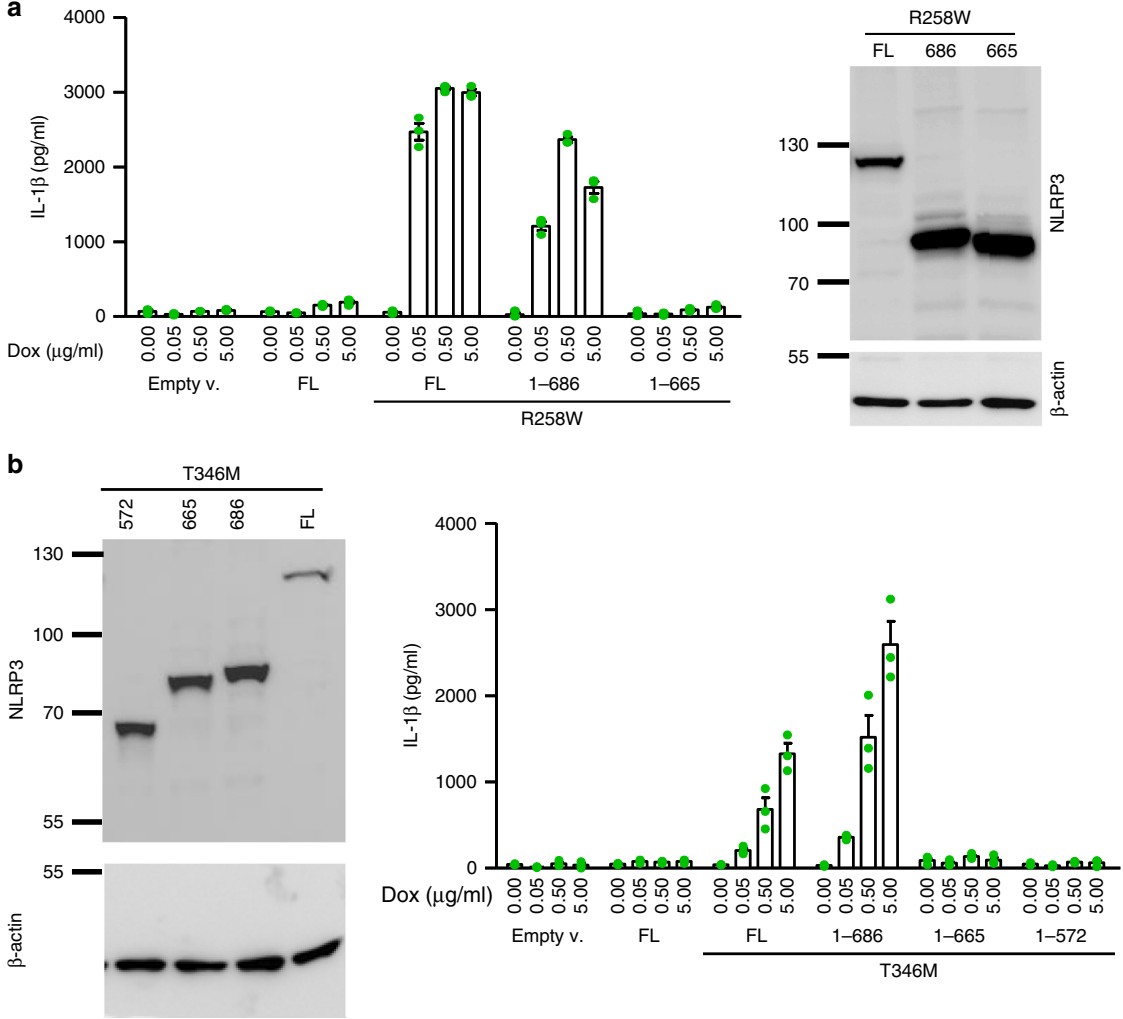

**Fig. 6** MiniNLRP3 (1–686) supports CAPS-associated constitutive activation. Truncated variants were introduced on a pathological substitution (**a**-R258W; **b**-T346M) background, and constitutive activation was followed by IL-1β release upon LPS priming. Representative of 3 (**a**) or 2 (**b**) independent experiments is shown. The mean and s.e.m. of 3 (**a**, **b**) biological replicates are shown

conformation and preventing NLRC4 oligomerization[17,19]. If similar stabilization with a presumably negligible effect of interactions with PYD is provided for NLRP3, then replacing the PYD for the CARD domains of either NLRC4 or ASC should result in NLRP3 trigger-dependent inflammasome assembly. To test this hypothesis, NLRP3-NLRC4 or NLRP3-ASC chimeric proteins or interaction domains (PYD, CARD) alone were expressed in $Nlrp3^{-/-}$ iBMDMs upon addition of doxycycline. While NLRP3 was active only in the presence of nigericin, replacement of the PYD for the CARD domain of NLRC4 or ASC resulted in the release of IL-1β in the absence of an NLRP3 trigger (Fig. 7a). As expected, no constitutive activation or nigericin-induced activation was observed for NLRP3 that lacked the PYD domain (Fig. 7a, Supplementary Fig. 14a, expression of those variants is shown in the Supplementary Fig. 14b). At high concentrations, the PYD and the CARD domains are prone to forming filaments[34,62]. However, no constitutive or trigger-specific cell activation was observed when isolated PYD or CARD domains were expressed (Fig. 7b), demonstrating that expression of these domains in the absence of an oligomerization domain is insufficient for activation under physiological conditions. Next, we were interested in whether CARD-containing chimeras can still engage ASC. ASC specks were observed with cells expressing CARD-

NLRP3 chimeras (Fig. 7c, Supplementary Fig. 14c). To test whether ASC is required for activation of CARD-NLRP3 chimeras, those variants and the ASC-GFP construct were introduced into $Asc^{-/-}$ iBMDMs. The introduction of ASC-GFP reconstituted the activation by nigericin (Supplementary Fig. 14d), while CARD-NLRP3 chimeras induced constitutive IL-1β release even in the absence of ASC (Fig. 7d, expression is shown in the Supplementary Fig. 14e). This phenotype is similar to that of NLRC4, which recruits ASC to form ASC specks but is partially functional even in the absence of ASC[63].

The truncation analysis demonstrated that the LRR is not important for stabilization of the inactive state, here, we additionally demonstrate that the interaction of the PYD with the rest of NLRP3, which cannot be accommodated by CARD domains, is crucial for stabilization of the inactive state.

**NLRP3 fails to induce NLRC4-like self-polymerization.** NLRC4 has a fascinating feature that upon activation with a bacterial ligand the ligand-NAIP complex engages additional NLRC4 molecules in a prion-like seeded polymerization event[20–22], resulting in inflammasome complexes where NLRC4 molecules greatly outnumber the NAIP/bacterial trigger complexes.

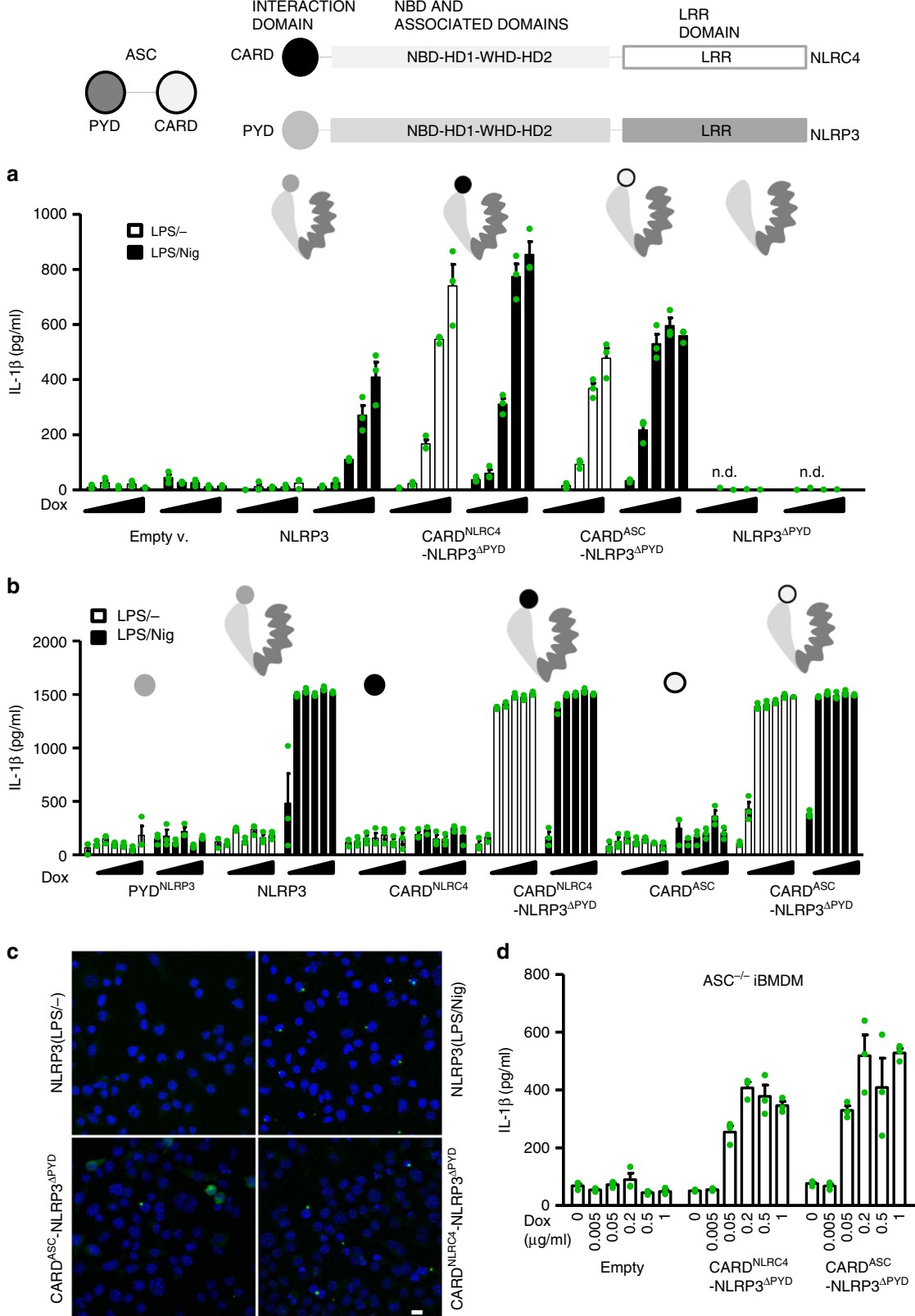

Apoptosomes, which are also caspase-activating platforms, self-assemble in a stoichiometric manner (reviewed in[19]). Thus far, it has been unknown whether NLRP3 is activated through the oligomerization of activated, multiple open conformation NLRP3 molecules (Fig. 8, I) or resembles NLRC4 amplification where one activated NLRP3 molecule could trigger oligomerization of

nonactivated NLRP3 molecules (Fig. 8, II). We reasoned that the constitutively active pathological variants are in the activated conformation and should be able to engage inactive wild-type NLRP3 molecules if the mechanism of activation resembles NLRC4's. If this is the case, a high response would be expected upon (co)expression of the low amount of the constitutively

**Fig. 7** Exchange of the PYD with the CARD domain leads to constitutive activation. **a** Chimeric CARD-NLRP3 variants and **b** corresponding interacting domains (PYD, CARD) were expressed in *Nlrp3*[−/−] iBMDMs by doxycycline (**a**: 0, 0.005, 0.01, 0.02, and 0.05 μg/ml; **b**: 0, 0.01, 0.05, 0.1, 0.5, 1, and 5 μg/ml) and primed with LPS (100 ng/ml) and then tested for their constitutive and nigericin-induced activation (5 μM, 1 h, note that 0 μg/ml dox is omitted in the latter case), as followed by IL-1β release. **c** For ASC speck labeling, cells were primed with doxycycline (0.5 μg/ml) and LPS (100 ng/ml) for 10 h after which the cells were fixed and labeled with an antibody against ASC. The formation of ASC specks in primed and nigericin-treated (1 h, 10 μM) cells harboring NLRP3 is shown as the control. The bar represents 10 μm. **d** CARD-NLRP3 chimeras were introduced into *Asc*[−/−] iBMDMs, and expression of the chimeras was induced with doxycycline. Constitutive activation was followed by measuring IL-1β release. Representative of 3 (**a**, **b**) or 2 (**c**, **d**) independent experiments is shown. The mean and the s.e.m. of 2 or 3 (**a**, **b**, **d**) biological replicates are shown. n.d. – below detection limit

active variant in the presence of excess wild-type NLRP3. Therefore, constitutively active pathological variants and full-length wild-type NLRP3 were expressed under the control of doxycycline in iBMDMs originating from wild-type mice, whereas endogenous NLRP3 expression was induced with LPS. To ensure that the protein levels of the pathological variants were low compared to those of endogenous NLRP3, the NLRP3 protein levels were estimated from western blots (Fig. 8a–d). The protein content was analyzed in the absence of LPS priming to provide basal endogenous NLRP3 levels and mutant expression upon the addition of doxycycline. The mutant expression is indeed low compared to LPS-induced expression of endogenous NLRP3 as doxycycline-induced increase of variant NLRP3 expression is only detected in the absence of LPS (Fig. 8c, d), while the increase is negligible when expression of endogenous NLRP3 is boosted by LPS. Using the same priming conditions, cells were tested for the release of IL-1β (Fig. 8e). In the absence of nigericin, only a minor release of IL-1β was observed in cell lines expressing constitutively active NLRP3 variants corresponding to activation of the pathological mutant (Fig. 8e). In the presence of nigericin, the release of IL-1β from macrophage lines expressing pathological variants or wild-type NLRP3 and endogenous NLRP3 was ten times higher than in the absence of the trigger (Fig. 8e), suggesting that pathological variants are unable to recruit wild-type NLRP3 in the absence of triggers. These results, therefore, argue against prion-like catalytic polymerization of NLRP3 and support apoptosome-like stoichiometric assembly as the likely scenario for NLRP3 activation.

## Discussion

This study provides new insight into the activation and regulation mechanisms of the NLRP3 inflammasome in biologically relevant cells at physiological levels of protein expression. Based on more than 20 variants of NLRP3, we demonstrated that the LRR domain is nonessential in NLRP3 inflammasome assembly. The NLRP3 truncation 1–686 fully phenocopies the full-length protein in terms of activation with several canonical NLRP3 inflammasome triggers, kinetics of activation, and pharmacological inhibition. As the nonfunctional 1–665 variant undergoes a characteristic drop in the BRET signal upon the addition of nigericin and retains ATP-binding propensity, the results suggest that this variant is able to sense trigger-released signals yet lacks the ability to assemble a functional inflammasome (Fig. 9a). The responsiveness of variants that lack the predicted LRRs emphasizes that the LRR domain does not harbor the sensing domain, which differs from the role of the LRR domain in Toll-like receptors and in plant NLR proteins[64,65]. The maintenance of full functionality of some variants by different activators is very strong evidence of the dispensability of LRRs for canonical inflammasome activation.

Interestingly, the LRR domain of the NAIP family members also does not determine the specificity for the bacterial ligands flagellin and T3SS proteins[23]. However, a suppressive role of the LRR domain was found in NAIPs[66] and NLRC4[17]. Previous studies have reported on the constitutive activation of NLRP3 lacking the LRR domain, which was much weaker than activation due to CAPS mutations[38]. Additionally, those results are likely due to the high protein expression when performed *in vitro*[34] or in the inflammasome reconstitution systems in yeast and HEK293 cells[13,37,67]. In contrast, mice expressing NLRP3, where the LRR domain was replaced by LacZ, did not show the CAPS inflammatory phenotype[68]. BMDMs from those mice had decreased activation of NLRP3 compared to BMDMs from wild-type mice and comparable to the *Nlrp3*[−/−] mice, but this could be due to the inhibitory effect of the large LacZ domain or due to the low expression of the fusion protein in the absence of the priming signal[68].

The present truncation study was primarily aimed at elucidating the mechanism of NLRP3 inflammasome activation. However, truncated NLRP3 variants have also been observed in nature, and splice variants lacking a part or the entire LRR domain have been observed in mice and in humans[69–71]. For example, a short transcript corresponding to amino acid residues 1–719 was identified in human blood cells[70]. During mouse mast cell maturation the predominant transcribed form is full-length NLRP3, three splice variants with splicing in the region that corresponds to the LRR were also detected[69]. One variant 1–829 is very similar to 1–823 (inflammasome-sufficient variant with three predicted LRRs) used in the present study. Enterovirus 71 proteases were shown to inactivate NLRP3 by cleavage at residues Q225 and G493[72], which is in agreement with presented results that variants equal to 1–665 or shorter are inactive in physiologically relevant conditions. A truncation variant resulting from nonsense mutation R554X was found in a patient with atypical autoinflammatory syndrome resembling familial Mediterranean fever with episodes triggered by exposure to cold[73]. This case study was published when NLRP3 was thought to be involved in the NF-κB pathway, and the effect of this variant on inflammasome activation has not been determined[73]. According to our results, it is likely that R554X by itself is incapable of forming an inflammasome, and the reason for the pathological phenotype that is different from typical CAPS should be investigated further. Recently, the first non-inflammasomal function of NLRP3 was reported. NLRP3 acts as transcription factor inducing the TH2 response[74], and this function depends on the LRR domain. Therefore, it is tempting to speculate that this may be the role of the LRRs, where inflammasome-sufficient but transcription regulation-deficient variants might drive a more proinflammatory phenotype than full-length NLRP3.

Activation of the NLRC4 inflammasome induced by bacterial activators that are sensed by members of the NAIP family has been well defined as structural information about the NLRC4 in the locked and activated forms is known[17,20,21,23,24]. However, the determined NLRC4 structures lack the interacting CARD domain whose functional equivalent in NLRP3 is a PYD domain. The constitutive activation of NLRP3/NLRC4 and NLRP3/ASC chimeras demonstrates that the PYD interacts with the rest of NLRP3 to repress the protein in an inactive conformation.

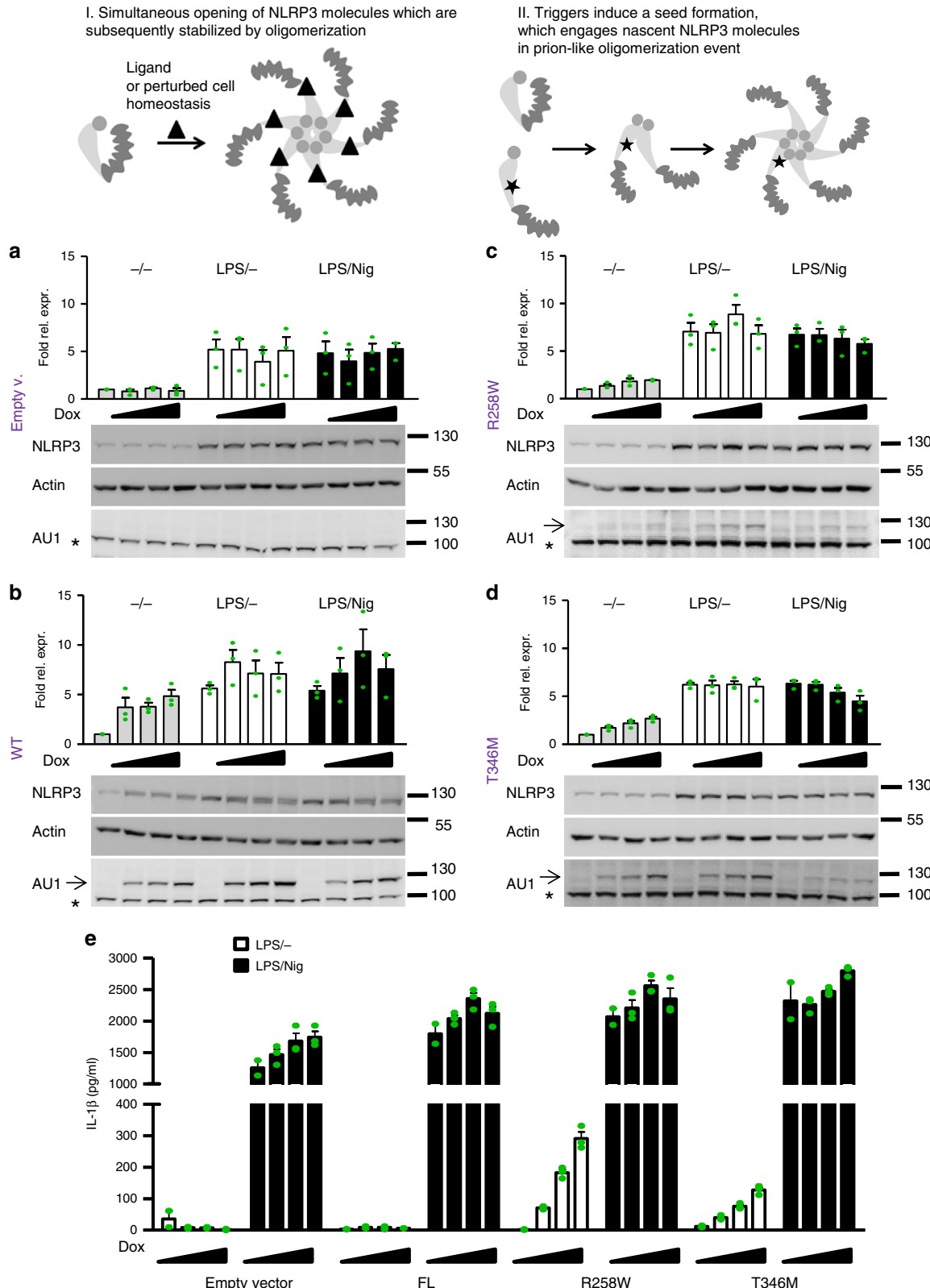

NLR proteins belong to the STAND family of AAA + ATPases, which include apoptosome-forming proteins such as Apaf-1. In contrast to NLRC4's self-oligomerization capability which amplifies the signal, Apaf-1 and Dark apoptosomes are activated as 1:1 complexes with their activators (cytochrome c for Apaf-1 and DroncCARD for Dark)[27–29]. Based on the present results, it is likely that NLRP3 is engaged in a similar type of complex. Our study provided evidence for the absence of the amplifying auto-catalytic assembly of the NLRP3 inflammasome, in contrast to the NLRC4 inflammasome (Fig. 9b). NLRP3 can be activated by a large number of different signals, either originating from patho-gens or based on sterile damage, while NLRC4 predominantly

**Fig. 8** Constitutively active variants fail to engage wild-type NLRP3 in seeded oligomerization. Two possible mechanisms of NLRP3 activation: Triggers induce simultaneous conformational change in NLRP3 molecules that assemble into oligomer (I), or triggers induce conformational change in a few NLRP3 molecules, which recruit nascent NLRP3 molecules to form an oligomer (II). **a–d** To estimate the protein levels of NLRP3, wild-type iBMDMs transduced with empty vector (**a**), wild-type (**b**), or R258W (**c**) or T346M (**d**) AU1-tagged NLRP3 variants were analyzed for NLRP3 content by Cryo-2 labeling in various testing conditions. NLRP3 levels were first normalized to actin and then to the basal endogenous NLRP3 level in the non-LPS and non-doxycycline-treated condition. Analysis of 3 biological replicates is provided in the charts. One blot per NLRP3 variant is shown. All blots used for analysis can be found in the Supplementary Fig. 18, arrow indicates the position of NLRP3 variants and * an unspecific band. **e** Activation of wild-type endogenous NLRP3 in wild-type iBMDMs was followed in the presence of constitutively active pathological variants. Cells were first treated with doxycycline (0, 0.5, 1, and 2 μg/ml) for 12 h, then with LPS (100 ng/ml) for 6 h, and afterward with nigericin or buffer (1 h). Representative of 3 (**e**) independent experiments is shown. The mean and the s.e.m. of 2 (0 μg/ml dox) or 3 biological replicates are shown

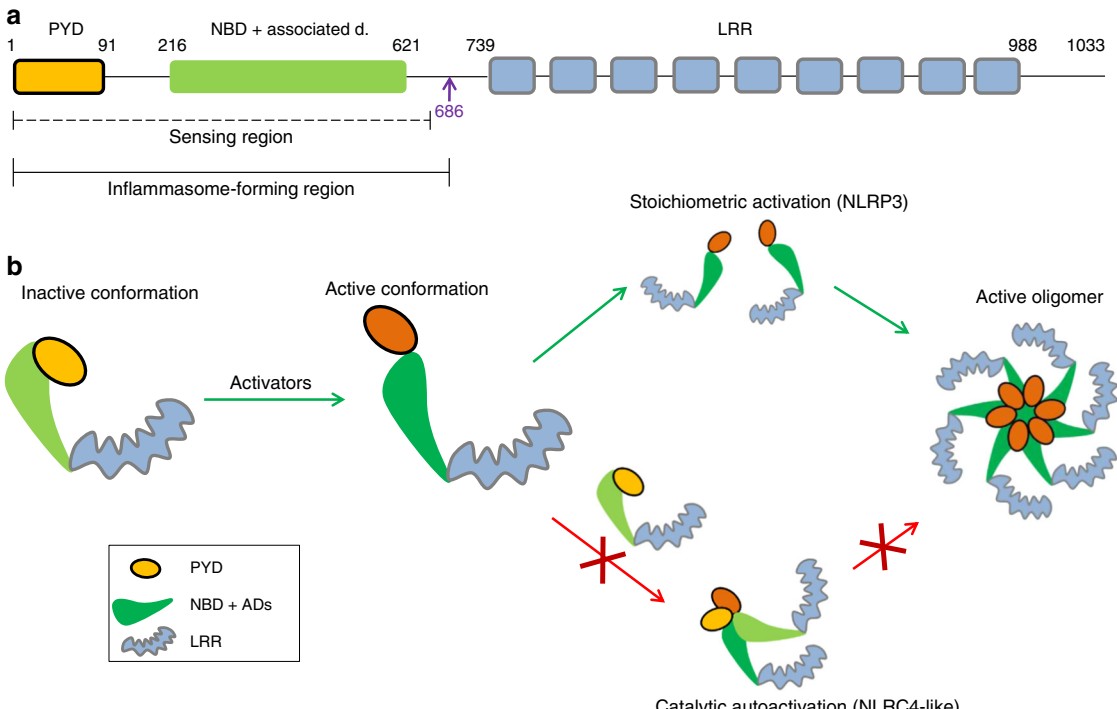

**Fig. 9** The role of NLRP3 domains in inflammasome autoregulation and activation. **a** The LRR domain is redundant for sensing and inflammasome formation and does not restrict the NLRP3 molecule in the inactive conformation. Instead, the sensing and inflammasome-forming regions are located N-terminally to the LRR domain, consisting of the PYD, the NBD, and associated domains and neighboring segments. **b** Before activation, NLRP3 is locked in the inactive form by the interactions of PYD with NLRP3 (92–665). Upon activation, inhibitory interactions are released, and several molecules of NLRP3 in the active conformation form an oligomer (green arrows). Unlike NLRC4, the activated NLRP3 molecule is unable to engage NLRP3 molecules in an inactive conformation (red arrows) in catalytic autoactivation, decreasing sensitivity and potential autoimmune activation by endogenous triggers

responds to structural patterns originating from pathogens (e.g., flagellin or type III secretion proteins). While the ability of NLRC4 to amplify the defensive response against infection is beneficial, the inability of NLRP3 to engage in prion-like polymerization may be important to prevent excessive activation by numerous triggers that occur under sterile inflammation and could trigger an autoinflammatory response.

The present study demonstrates that the NLRP3 self-inhibition and oligomerization mechanisms diverge from those of NLRC4, manifesting the versatility of tuning the appropriate response by innate immunity sensors for the recognition of invading and self-derived signals.

## Methods

**Materials**. The sources of the chemicals and other materials used are listed in Supplementary Table 1.

**Mice**. All animal experiments were performed according to the EU 2010/63 directive and were approved by the Administration of the Republic of Slovenia for Food Safety, Veterinary and Plant Protection of the Ministry of Agriculture, Forestry and Foods, Republic of Slovenia (Permit Number U34401–4/2017/4 for the described animal experiment and U34401–36/2015/2 for tissue collection). *Nlrp3*-deficient mice C57BL/6-*Nlrp3*[tm1Tsc75] on a C57BL/6 background, a kind gift from Dr. Isabelle Couillin (Experimental and Molecular Immunology and Neurogenetics, University of Orleans, Orleans, France), were housed in the SFP animal facility in an IVC system. Littermates of the same sex were randomly assigned to experimental groups (3 or 4 mice per group). No particular randomization method was used to assign animals to experimental groups. No power calculations were performed to choose group size. The mice were fed standard chow and offered fresh tap water ad libitum.

Eight- to 12-week-old male and female mice weighing around 25 grams, with normal health and immune status, were used for the experiments.

Before implantation, *Nlrp3*[−/−] iBMDMs stably encoding empty vector, wild-type NLRP3, 1–686, or 1–665 were primed with 100 ng/ml LPS and 1 μg/ml doxycycline for 12 h. Cells were detached and washed several times with PBS. Detachment of LPS-primed cells either through scraping or by trypsin induced inflammasome activation and the addition of another trigger (e.g., nigericin or silica) had no further effect. LPS priming ex vivo was used as LPS enhanced

transgene expression, which is not specific to NLRP3 variants, as the same effect was observed with fire-fly luciferase.

Mice were intraperitoneally injected with $2 \times 10^6$ appropriate cells using a 29 G needle (Beckton Dickinson). After 2 h (for cytokine determination) or 24 h (for neutrophil infiltration), the mice were humanely sacrificed with $CO_2$ asphyxiation, and peritoneal lavage was performed. Peritoneal cells were obtained for further flow cytometry analysis to determine infiltration of Gr-1-positive cells. The investigator was not blinded during the experiment or when assessing the outcome.

For preparation of BMDMs, C57BL/6 mice were sacrificed. Bone marrow was flushed from the femurs and tibias, and BMDMs were prepared with cultivation of the bone marrow in RPMI 1640 medium supplemented with 20% FBS, 40 ng/ml M-CSF, and penicillin/streptomycin for 6 days. Then the cells were seeded and primed with LPS (100 ng/ml for 6 h) to follow endogenous NLRP3 expression.

**Cell lines**. Immortalized bone marrow–derived macrophages (iBMDMs) from wild-type C57BL/6 mice and $Asc^{-/-}$ and $Nlrp3^{-/-}$ mice were a gift from Prof. K. A. Fitzgerald (University of Massachusetts Medical School, Worcester, MA, USA)[7]. The iBMDMs were grown in DMEM supplemented with 10% FBS (Gibco, Invitrogen) in a humidified $CO_2$ incubator with 5% $CO_2$ at 37 °C. Retrovirus for transduction was produced in packaging cell lines Platinum-GP (Cell Biolabs, RV-103), Gryphon Eco (Alelle Biotechnology, ABP-RVC-10002), and Gryphon Ampho (Alelle Biotechnology, ABP-RVC-10001), which were cultivated in the same growth conditions as the iBMDMs. The HEK293 cell line (ATCC, CRL-1573) was also grown in DMEM supplemented with 10% FBS. All cell lines were tested mycoplasma-negative and were not authenticated. The HEK293 cell line is commonly used in molecular biological studies as it is easily genetically manipulated.

**Construct preparation**. For inducible expression upon retroviral transduction, all constructs were inserted into the pRETROX Tre3G plasmid (Clontech). The mouse *Nlrp3* encoding synthetic gene was ordered from GeneArt (Life Technologies). Truncated mouse *Nlrp3* variants were prepared with PCR using Phusion HF polymerase (Thermo Fisher Scientific) using primers (Supplementary Table 2: O1–O22, O61) and inserted into pRETRO X Tre3G at BamHI/EcoRI sites. Human *NLRP3* and *NLRC4* genes were obtained from Invivogen. Truncation human *NLRP3* mutants were prepared with PCR using Phusion HF polymerase (primers: O23–O26) and inserted into pRETRO X Tre3G at the BamHI/NotI sites. Point mutations were introduced by site-directed mutagenesis using the O45–O60 primers and the Phusion HF polymerase. Chimeric constructs of mouse NLRP3 and human NLRC4 or human ASC (primers: O34–O44) and BRET constructs (primers: O28–O33) were prepared with PCR ligation. BRET constructs were further subcloned into pcDNA3 using the BamHI/NotI sites. pMXs-puro(Cell Biolabs)/ASC-GFP was used for control transduction of $Asc^{-/-}$ iBMDMs. All constructs were verified with DNA sequencing (GATC). The plasmid encoding *hASC* was a kind gift from K.A. Fitzgerald[76].

**Transduction**. The Tet-On retroviral system (Clontech, 631188) was used for inducible expression of NLRP3 variants[36]. The doxycycline-inducible system was used because we observed that expression of some truncated variants was lost when the gene was under the control of a constitutive promoter. Immortalized BMDMs from $Nlrp3^{-/-}$, $Asc^{-/-}$ and wild-type mice (seeded a day before transduction at $3–4*10^5$ cells/well of 6-well plate) were transduced with the Tet-On 3G transactivator retrovirus, produced after transfection of Platinum-GP (Cell Biolabs) packaging cells (seeded a day before transfection at $2*10^6$ cells/well of 6-well plate) with 1.5 μg pCMV-VSV-G (Cell Biolabs) and 2.5 μg pRETROX Tet3G (Clontech) using 10 μL of Lipofectamine 2000. Transduced cell weres selected using G418 (1.5 mg/ml). Single-clone cell lines were used for further experiments and transduced with NLRP3 variants or empty vector encoding retroviruses, produced by transfection of the 4 μg pRETROX Tre3G plasmids (Clontech) using 10 μL of Lipofectamine 2000 into Gryphon Ampho or Eco (Alelle Biotech) cells. 2–3 days after transduction, the cells were selected by growth in puromycin (6 μg/mL) and G418 (1.5 mg/ml) in DMEM supplemented with 10% FBS.

**Cell stimulation and NLRP3 inflammasome activation**. Cells were seeded at $1.0–1.5 \times 10^5$ cells per well of 96-well (for ELISA, LDH assays) in the morning in DMEM supplemented with 10% FBS. All stimulation experiments were performed in serum-free DMEM. In the evening, the cells were primed with a combination of ultra-pure LPS (100 ng/mL) and doxycycline for the designated time. After priming, the medium was exchanged for the medium or stimulation buffer (10 mM Hepes, pH 7.45, 147 mM NaCl, 13 mM D-glucose, 2 mM KCl, 2 mM $CaCl_2$, and 1 mM $MgCl_2$) containing the activators nigericin (Sigma), alum (Thermo), silica (Invivogen), and imiquimod (Invivogen). After a designated activation period, the supernatants were collected and analyzed for IL-1β, IL-18, and LDH. In the inhibition experiments, the inhibitors were added before the addition of the activators.

**LDH assay**. For the LDH assay, the supernatants were analyzed for the presence of LDH activity using the Promega Viability Assay, and absorbance at 492 nm was measured using the multiplate reader SinergyMx (BioTek) and Gen 5.1.10 software (Biotek). Supernatant from 0.1% Triton X-100−treated cells was used as positive

control. The percentage of the LDH release was calculated using the supernatant of the untreated cells as negative control and Triton X-100-treated supernatant as 100% LDH release.

**ELISA**. The concentrations of the secreted IL-1β, TNF-α, IL-6, and IL-18 were measured with ELISA (e-Bioscience) according to the manufacturer's instructions. For measuring the cytokines in peritoneal lavage, cell-free lavage was concentrated, cytokine concentrations were measured with ELISA, and amount of cytokines was calculated taking into account the dilution factor and volume of the lavage.

**ASC speck formation by immunofluorescence**. $10^5$ cells were seeded onto poly-L-lysine-coated coverslips (Corning). Cells were primed (100 ng/ml LPS, 1 μg/ml doxycycline) overnight. The medium was then replaced with nigericin in a stimulation buffer. After 45 min of treatment with nigericin, the cells were fixed with paraformaldehyde (4 %, Electron Microscopy Sciences) for 15 min and permeabilized with 0.2% saponin and 1% BSA in PBS for 30 min[59]. Further, the cells were incubated with a primary antibody against ASC (BioLegend, 653902, 1:500) for 1 h at room temperature and after washing, in secondary Alexa Fluor 488 donkey anti-mouse IgG (Thermo Fisher Scientific, A21202, 1:200) in a permeabilization/blocking solution. Phalloidin-CF647 (Santa Cruz) was added for 30 min, and then, the coverslips were washed with PBS and before mounting briefly with distilled water. Coverslips were mounted on slides with Prolong Diamond Antifade solution with DAPI (Invitrogen). Images were acquired either with a Nikon Eclipse Ti microscope with a 60X Plan Apo Vc objective (numerical aperture, 1.40) and a digital Sight DS-QiMc camera (Nikon) with Z optical spacing of 0.2 μm and 387-nm/447-nm, 472-nm/520-nm, and 650-nm/668-nm filter sets (Semrock) or with Leica TCS SP5 laser scanning microscope mounted on a Leica DMI 6000 CS inverted microscope (Leica Microsystems, Germany). In the latter case, a 405 nm laser line of 20 mW diode laser was used for DAPI excitation and emitted light was detected between 415 and 450 nm. A 488 nm laser line of 100 mW argon laser with 10% laser power was used for detection of Alexa 488 fluorescence, where emitted light was detected between 500 and 600 nm. Maximum-intensity projection of images was achieved with NIS-Elements AR software (Nikon, version 4.30.02) or Leica LAS AF software (version 2.7.2.9586).

**Western blotting**. For detection of NLRP3 protein variants, cells in a 48-, 24-, or 6-well format were washed twice with cold PBS and lysed. The protein concentration in the cell lysate was measured with BCA. For detection of cleaved caspase-1 and IL-1β, after stimulation, the cell supernatants were concentrated with 10k Amicon Ultra centrifugal filters (Millipore, Merck). Proteins were separated on SDS-PAGE gels, blotted onto the nitrocellulose membrane (GE Healthcare), and detected with appropriate primary and secondary antibodies for the detection of caspase-1 p20 (Casper-1, Adipogen, AG-20B-0042-C100, 1:1000) followed by HRP-conjugated goat polyclonals to rabbit IgG (Abcam, ab6721, 1:3000). IL-1β was detected with the Genetex antibody (1G-GTX74034–100, 1:2000). NLRP3 was detected using Cryo-2 (Adipogen, AG-20B-0014-C100, 1:2000) as the primary antibody and anti-mouse HRP-conjugated antibodies (Jackson ImmunoResearch, 115–035–003, 1:3000 or GE Healthcare, NA931, 1:1000). SuperSignal West Pico, Femto Chemiluminescent Substrate (Thermo Scientific), and ECL (Amersham, GE Healthcare Life Sciences) were used for detection of HRP-labeled bands with G-box (Syngene) using Genesnap 7.09 software.

Uncropped images of blots are available in Supplementary Figs. 15–23.

**Immunoprecipitation**. pcDNA3-N-HA-NEK7 was a gift from Bruce Beutler (Addgene plasmid # 75142)[35]. C-terminally FLAG-tagged NLRP3 variants in pcDNA3 (2 μg) were co-transfected with the NEK7 plasmid (2 μg) into the HEK293 cells (seeded a day before at $1.8 \times 10^6$/well of a 6-well plate) using Lipofectamine 2000 (10 μL per reaction) (Invitrogen). An empty pcDNA3 vector was used as negative control. Two days after transfection, the cells were washed with PBS and lysed in 50 mM Na-phosphate buffer, pH 7.5, 150 mM NaCl, 0.5% NP-40, protease inhibitor cocktail (Roche). Cells were lysed through a 26 G needle several times. One-fifth of the sample was used as loading control, and the rest was incubated with antibodies against FLAG (Sigma F7425, 2 μL per reaction) overnight at 4 °C and then with Dynabeads Protein G (Thermo Fisher Scientific, 25 μL per reaction). NEK7 was detected with antibodies against HA-tag (Invivogen, abhatag, 1:667). For ASC or NEK7 immunoprecipitation from BMDMs, the cells were incubated with pan-caspase inhibitor Z-VAD-FMK (Invivogen) 30 min before nigericin addition. After nigericin stimulation cells were lysed in 50 mM Tris-HCl, 150 mM NaCl, 2% Triton X-100, pH 8.0. Then 500 μg of total proteins were incubated with anti-GFP antibody (Invitrogen, A11122, 5 μg/IP sample) to capture YFP-tagged NLRP3 variants and Dynabeads Protein G (Thermo Fisher Scientific, 25 μL/reaction) at 4 °C overnight. ASC was detected with AL177(Adipogen, AG-25B-0006, 1:1000), and NEK7 with an antibody from Abcam (ab133514, 1:2000). ASC was after co-IP detected with Protein A-HRP (Abcam, ab7456, 1:1000).

**ATP agarose pulldown**. NLRP3 variants were overexpressed in HEK293 cells. Two days after transfection, the cells were lysed in 10 mM Na-phosphate buffer, 150 mM NaCl, 0.1% NP-40, pH 7.3. Then 150 μg of total proteins were incubated

overnight at 4 °C with ATP beads (ATP Separopore 4B-CL, bio-World) (in the presence or absence of 40 µM CY-09) or control beads (Separopore 4B-CL, bio-World) in 10 mM Na-phosphate buffer, 150 mM NaCl, 0.01% NP-40, pH 7.3. After several washes, NLRP3 variants were eluted in SDS sample buffer and detected with western blot with the Cryo-2 antibody.

**BRET**. HEK293 cells were transfected with plasmids encoding double-tagged NLRP3 (intramolecular BRET; YFP-NLRP3-rLUC) or pairs of single-tagged human NLRP3 variants (intermolecular BRET) and rLUC-NLRP3 variants only. Titration was performed by transfection of different amounts of plasmids encoding YFP-NLRP3-rLUC or NLRP3-rLUC variants (pCDNA 3.1 was added to enable the transfection of equal amounts of total DNA in all cases). One day after transfection, the cells were plated into poly-L-Lysine-coated 96-well white plates. The next day, the cells were washed, and luminescence reading was performed with Synergy Mx (Biotek) (rLUC filter, 485 ± 20 nm; and YFP filter, 530 ± 25 nm) at 37 °C in 10 mM Hepes, pH 7.45, 147 mM NaCl, 13 mM D-glucose, 2 mM KCl, 2 mM $CaCl_2$, and 1 mM $MgCl_2$. The BRET ratio was defined as the difference of the emission ratio 530 nm/485 nm of co-transfected rLUC and YFP fusion proteins and the emission ratio 530 nm/485 nm of the rLUC fusion protein alone[54,61]. Results were expressed in milliBRET units (mBU). All data points for BRET experiments are shown in Supplementary Fig. 24.

**Flow cytometry**. To determine the number of infiltrated neutrophils in the peritoneal lavage, the cells were first fixed with Rotihistofix (Roth) at 4 °C for 30 min and washed several times in PBS + 3% FBS. Cells were labeled with Gr-1-FITC (1.5 µL RB6–8C5, Thermo Fisher Scientific, 11–5931–82) and F4/80 PerCP-Cy5.5 (1.5 µL BM8, Thermo Fisher Scientific, 45–4801–82) for at least 30 min on ice, washed several times, and followed with the flow cytometer CyFlow Space (Partec) using FloMax software (v. 2.70, Quantum Analysis group), where the FL1 and FL3 channels were used to discriminate between different cell populations (neutrophils and macrophages). The number of cells/ml was determined and the total number of Gr-1+, F4/80− cells was calculated taking into account the sample volume and the percentage of cells in the defined region. Example plots can be found in Supplementary Fig. 25.

**Confocal microscopy**. To follow the localization of the NLRP3 variants, stable Nlrp3−/− cell lines encoding NLRP3-YFP (full-length mouse NLRP), (1–665)-YFP, or (1–686)-YFP cells were seeded into a µ-slide (Ibidi) ($3 × 10^5$ per well). Cells were non-treated or treated with LPS (100 ng/ml) and doxycycline (1 µg/ml) overnight. In the morning, the cells were washed and incubated with Hoechst (1 µg/ml; Immunochemistry Technologies) for 5 min at 37 °C, and washed several times before nigericin was added (5 µM). After 1 h, the cells were observed under a Leica TCS SP5 laser scanning microscope mounted on a Leica DMI 6000 CS inverted microscope (Leica Microsystems, Germany) with an HCX plan apo 63 × (NA 1.4) oil immersion objective used for imaging. A 405-nm laser line of a 20-mW diode laser was used for Hoechst excitation, and emitted light was detected between 450 and 510 nm. A 514-nm laser line of an 100-mW argon laser with 25% laser power was used to detect YFP, where emitted light was detected between 530 and 600 nm. To acquire and process the images, Leica LAS AF software was used.

**Statistical analysis**. The sample size was chosen depending on the experiment used and based on our previous experience to provide an adequate power. An unpaired two-tailed t-test was used for pairwise comparison. When variances were not similar between the groups, then unpaired t-test with Welch correction was used. Statistical tests were calculated with GraphPad Prism 5 (GraphPad Software, version 5.01).

**Data analysis**. Microscopy images were processed in ImageJ (National Institutes of Health, version 1.50i). All images within the same experiment were processed in the same way. Quantification of the protein levels was performed with ImageJ. Band intensity was first normalized to the loading control, and then to the normalized reference (defined in the figure legends). Flow Cytometry data were processed with FlowJo (FlowJo, version 10). Data from ELISA, LDH and BRET measurements were processed in Excel (Microsoft, version 2010), Origin (Origin Lab, version 8.1.34.90) and GraphPad Prism 5 (GraphPad Software, version 5.01).

**Reporting Summary**. Further information on research design is available in the Nature Research Reporting Summary linked to this article.

## Data availability
A Reporting Summary for this Article is available as a Supplementary Information file. All other data supporting the findings of this study are available from the corresponding authors on reasonable request.

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

## Acknowledgements

We are grateful to L. Brvar, J. Lenarčič, R. Bremšak, and I. Škraba (National Institute of Chemistry, Slovenia), M.C. Baños and A. I. Sánchez (IMIB-Arrixaca, Spain), and A. Perčič and K. Skulj (University of Ljubljana, Biotechnical Faculty, Slovenia) for technical assistance. The authors would like to thank Prof. K. A. Fitzgerald (University of Massachusetts Medical School, USA) for providing immortalized mouse macrophages and plasmid containing ASC, Dr. I. Couillin (University of Orleans, France) for providing the *Nlrp3*⁻/⁻ mice, Prof. B. Beutler (University of Texas Southwestern Medical Center, USA) for providing the NEK7 plasmid, Dr. F. Martin Sanchez (IMIB-Arrixaca, Spain) for valuable advice on labeling ASC specks, Dr. Carlos de Torre Minguela (IMIB-Arrixaca, Spain) for advice on immunoprecipitation, and Dr. M. Benčina (National Institute of Chemistry, Slovenia) for help with confocal microscopy. The authors are grateful to Prof. S. Horvat (University of Ljubljana, Biotechnical Faculty, Slovenia) for advice on the animal study. This work was funded by Slovenian Research Agency project grants J3–5503 and J3–6791 to I.H.-B., research core funding P4–176 to R.J., and young researcher's PhD grants to P.S and L.K. This

work was supported by grants from Instituto Salud Carlos III–Fondo Europeo de Desarrollo Regional (PI13/00174 to P.P.), Sysmex (to P.P.), and the European Research Council (ERC-2013-CoG 614578 to P.P.). I.H.-B. and P.P. would like to acknowledge the networking support by the COST Action BM-1406. This work was supported by an STSM Grant from COST Action BM-1406 (to I.H.-B.). The funding agencies had no role in the study design and interpretation of results. K.C. is a student of the Uniform Second-level Master's Program of Medicine at the Medical Faculty, University of Ljubljana.

## Author contributions

I.H.-B. designed the study, performed the experiments, analyzed the results, and wrote the manuscript. D.L. performed the animal experiments. P.S., K.C., L.K., D.A.-B., and A.T.-A. performed the experiments. P.P. provided valuable suggestions, discussed the results, and wrote the manuscript. R.J. designed the study, provided valuable suggestions, discussed the results, and wrote the manuscript.

## Additional information

**Competing interests:** The authors declare no competing interests.

