## [Peer Review File · Nature Communications]

Reviewers' comments:

Reviewer #1 (Remarks to the Author):

Nod-like receptor (NLR) family, pyrin domain containing 3 (NLRP3) is involved in the early stages of inflammatory responses where it senses cellular damage or distress due to bacterial infection. In the manuscript, the authors have constructed more than 20 truncation variants of murine NLRP3, established their stable expression system in NLRP3 KO BMDMs (Tet-on system), and they identified that leucine-rich repeat (LRR) is not essential for the activation of NLRP3 inflammasome. Regarding their conclusion, the authors also have tested a LRR deletion mutant (called mini-NLRP3) act as the minimal responsive NLRP3 in the experimental conditions. Overall of my impression is that demonstrating absence or dispensable is a daunting task. As they mentioned in the introduction section (NLRP3 inflammasome is activated by a wide range of DAMPs), their experimental conditions might be a limited range of monitoring (mostly using nigericin and provided data were preliminary). And most importantly, a functional role and /or physiologic relevance of LRR were not identified/demonstrated in the manuscript. For example, how the mini-NLRP3 sense DAMPs without LRR (molecular mechanism)?

These are specific comments for the authors which must be addressed.

- 1). BRET experiments: What does the spike (decreasing BRET signal after adding nigericin) means? If the nigericin sensing site in NLRP3 have already identified based on crystal structure, it would be great to make a non-sensing NLRP3 mutant and test the BRET experiment.
- 2). Immunofluorescence images: In Figure 3C, the represented images were less resolution. And also, some positive controls (WT BMDMs?) must be provided.
- 3). NLRP3 mutants (Cys mutants, T346M, R258W): It was unclear for me that how those mutants fit into the story of the activation of NLRP3 inflammasomes.
- 4). Physiologic relevance of LRR in NLRP3: It is interesting to know that a phenotypic role of natural truncated version of NLRP3. To test this question, your system (knock in mice; mini-NLRP3) would be a great advantage.
- 5). Supplementary figure: Full blot images (western blots) in the manuscript were missing.

Reviewer #2 (Remarks to the Author):

In this study Hafner-Bratkovic and colleagues address potential mechanism of NLRP3 inflammasome activation and oligomerization by employing NLRP3 truncation and chimeric variants in reconstituted macrophages to mimic endogenous levels. The authors show that the C-terminal LRR domain of NLRP3 is dispensable for activation and self-regulation, while the N-terminal pyrin domain is required to maintain NLRP3 in the inactive conformation. The authors also demonstrate that unlike NLRC4, active NLRP3 does not engage wildtype molecules to oligomerize and such lack of signal amplification was reasoned to be a protective mechanism to inhibit auto-inflammatory conditions.

The studies presented here are well designed and potentially provide novel insights into NLRP3 function that are significantly different from the current paradigm. However the approach of assessing truncation mutations has limitations and some type of complementary approach assessing endogenous proteins would improve the study. I do however have a number of concerns:

- 1) In Figure 1 the authors reason loss of NLRP3 activity to low protein expression/degradation of truncation variants 1-850 to 1-996. Although diminished protein is observed in Figure 1D there is still substantial expression of these truncation variants. Hence it does not make sense that no IL-1beta secretion is measured from the truncation variants 1-850 to 1-996. The authors need to perform additional studies to define the reason for this defect. These data also raise the possibility that LRR domains confer stability to NLRP3.
- 2) In Figure 1B a number of the truncation mutants induce ~700-800 pg/ml of IL-1beta with LPS priming suggesting that they may be driving spontaneous activation of the NLRP3 inflammasome. In addition the length of priming (11 hours) is quite long. Have the authors tried these experiments with a shorter priming time (4-6 hours)?
- 3) Throughout the manuscript it would be helpful to include control cytokines such as TNF and IL6 to show the affect of the truncation mutations are specifically on NLRP3 inflammasome and IL-1beta.
- 4) In Figure 4A it would be helpful to provide additional supportive data such as the levels of proinflammatory cytokines in the peritoneal lavage including IL-1beta levels. Also in the experimental condition used where is the NLRP3 activating signal coming from?
- 5) In Figure 4B data the scale bars in each panel are of varying size - all panels should be the same magnification.
- 6) Figure 5 - the authors should consider performing co-ip panel with 1-665, 1-688, and FL NLRP3 variants assessing other inflammasome components (ASC, caspase-1) involved in complex formation in addition to NEK7.
- 7) The axis label for Figure 5E has been cut off.

Reviewer #3 (Remarks to the Author):

The authors present a study on the molecular mechanisms of NLRP3 activation. This is an extremely exciting and much debated area in the field of inflammasomes and inflammation. A key finding is that the LRR domain of NLRP3 does not appear to function as a sensor domain or auto-inhibitory domain, as a truncation mutant lacking the entire LRR domain is able to reconstitute NLRP3-deficient cells with the same phenotype as full-length NLRP3. This is an interesting and novel observation, which will influence thinking in the field. Further truncations are performed and the minimal length of NLRP3 that retains function is found to be NLRP3 1-686 or 'mini-NLRP3'. This 'mini-NLRP3' is defined as 1-688 or human NLRP3 (when expressed in mouse cells). A number of cysteine residues in NLRP3, that may be expected to be involved in redox regulation, were mutated but did not result in any decrease in NLRP3-dependent IL-1 β secretion in this system. Surprisingly, 'mini-NLRP3' was able to interact with NEK7 in an overexpression system. Experiments using intramolecular and intermolecular BRET found that 'Mini-NLRP3' was also able to interact with itself and full-length NLRP3 when triggered by nigericin. Finally, the authors propose that the PYD domain of NLRP3 is inhibitory and maintains NLRP3 in an inactive conformation as replacing NLRP3 PYD with a CARD domain renders the protein hyperactive in a manner similar to CAPS mutations. The authors extensively exploit the doxycycline inducible retroviral expression system that they have developed to make some interesting and challenging observations on the way NLRP3 is activated.

Overall assessment: The manuscript has some interesting observations about NLRP3 signalling/function, but the manuscript as it currently stands really lacks clarity and flow. There are general issues with language and continuity, the rationale for particular experiments is missing, and the manuscript figures are extremely difficult to follow (many are missing labels, legends etc so the reader often has to guess what the data is). Many of the manuscripts figures require additional experiments to justify the author's claims. This manuscript would only be suitable for publication in Nat Comms if the authors were willing and able to perform really substantial

revisions to bring the manuscript up to the required quality.

Major comments:

This study makes extensive use of truncation mutants to map domains required for NLRP3 activation and function. Such experiments, while extremely useful, need to be performed and interpreted with caution – mutation can have small effects on NLRP3 protein expression that can have a large effect on NLRP3 function (as expression level is critical for NLRP3 signalling), and mutations may have unintended consequences on gross protein structure. While the former (expression level) is somewhat addressed, the latter is not addressed at all. Can the authors be sure that non-functional NLRP3 truncation mutants are folded correctly (particularly in the NACHT domain that is critical for signalling)?

Figure 1: The retroviral expression system seems to be appropriate for studying the role of the LRR domain in the regulation of NLRP3. While the results clearly show that truncation of the whole LRR domain (731) does not result in loss of function, the results with the truncations from 850-996 are puzzling, and cannot be simply explained away by a relatively small decrease in the level of protein expression. These expression levels would need to be quantified (relative to their loading controls) perhaps in comparison to the level of WT FL NLRP3 expression induced by different levels of doxycycline as presented in Supplemental Figure 2 C and D. What is the minimal level of expression required for NLRP3 to function in this system? Are these truncations (850-996) expressed below this level? Could one of the truncations (850-996) be expressed at a higher level to demonstrate that it is still functional? This is a fundamental observation in the study and these issues need to be addressed.

Line 176 and Supplemental Figure S4: As the expression of all the NLRP3 variants is under the control of doxycycline there is no relevant transcriptional regulation of NLRP3 as these are NLRP3-deficient cells. Post-transcriptional priming of NLRP3 is a rapid event (within 5 mins of LPS treatment) so I don't see any relevance to looking at long LPS priming times. To address whether 'mini-NLRP3' is post-transcriptionally regulated in a similar manner to full length NLRP3 an experiment should be performed using short priming times with LPS (e.g. 0-30 min) and nigericin stimulation. Cell death and/or caspase-1 activation can then be measured as a readout of NLRP3 activation. As 'mini-NLRP3' lacks numerous sites for post-translational modification that have been described in the literature (e.g. PTPN22, PKA, BRCC3) this is something that needs to be addressed.

Figure 2: Can you comment on what domains/features are present in NLRP3 686 (minimally responsive) and NLRP3 665 and NLRP3 572 (Supp fig S3A) that fail to respond?

Figure 3C Although it is clear that NLRP3 1-667 does not induce ASC speck formation in response to LPS/Nigericin it is hard to determine whether the levels of ASC expression are similar in these different cell lines. This should be addressed with western blot or quantification of microscopy data. Some of the panels don't appear to be in focus (eg. 1-688 LPS). Although NLRP3 interacts with ASC via its PYD domain it would be interesting to examine whether the NLRP3 truncations presented here interact with ASC to the same extent using for example a co-immunoprecipitation assay.

Figure 4A I am unfamiliar with this model of peritonitis – could some reference be provided for previous examples of its use? What statistical test was performed?

Figure 6A: As the CARD domain of NLRC4 and ASC can interact directly with caspase-1 it is possible that expression of these chimeric proteins can trigger caspase-1 autocatalytic processing and activation. The formation of ASC specks in these cells needs to be assessed. An important control here would be to express NLRP3 with no PYD domain (rather than with the PYD domain

replaced with a CARD).

It would be very interesting to truncate NLRP3 from the N-terminus to assess which elements of the PYD domain are required to maintain the inhibited state of NLRP3.

Figure 6B This is an intriguing experiment and although it hints at the stoichiometric model of NLRP3 oligomerisation there are a number of caveats. It is not really definitively known that the point mutations used are in an 'open conformation'. This data needs to be accompanied by data showing the expression level of NLRP3 in each case, or the results cannot be interpreted.

The ATPase activity of NLRP3 is known to be essential to its function. Is it possible that these mutations have affected the nucleotide binding/hydrolysis ability of NLRP3?

Minor Comments:

Line 1-2 The title is extremely unclear and should be changed to something that more clearly indicates the experimental results.

Line 28 'self-encoded' is not correct (e.g. ATP is not an 'encoded' molecule), 'self-derived' is more appropriate.

Line 29 and Line 32 The Leucine-rich... the use of the definite article throughout the manuscript needs to be checked.

Line 43-45 The list of abbreviations is incomplete. Some that are missing include: DAMPs, PAMPs, BIR, LRR.

Line 73 NEK7 is not regarded as an NLRP3 'ligand' – it's a protein that interacts with NLRP3 and is required for NLRP3 function.

Line 78 This phrasing implies that HET-E and TP-1 are NLR proteins which they aren't.

Line 96 What does an 'inflammasome particle' refer to? Please define this or be more specific.

Line 110 'LRR-deficient minimal NLRP3' is quite a confusing term. Perhaps something like NLRP3 without the LRR domain would be clearer?

Line 125 what does 'available data' refer to? Is this published work?

Figure S1 Needs much clearer labelling. For example, in the NCBI schematic what is the F beside NBD? In the Albrecht model what are 1-2-3 to the right of the NBD? In the Proell model what do C-WH and SH2 refer to? These are not defined anywhere.

Figure 1 A What does AD mean? Perhaps this figure would be clearer if the LRR domains were first defined in a consistent manner and then numbered 1-9.

Figure 1 B-C For only two biological replicates – presenting the mean +/- range rather than standard deviation may be more appropriate.

In these bar charts and throughout the rest of the figures it is never indicated what the different coloured bars refer to – I can only assume that the white bars here are LPS primed cells? For a reader to be able to interpret this data it really needs to be labelled.

Figure 1D Label the blot for NLRP3.

Supplemental Figure 2A – NLRP3 deficient cells are conventionally termed Nlrp3^{-/-} and this should

be used in the manuscript.

Supplemental Figure 2B – Why is luciferase activity being measured here? Does the NLRP3 expression construct also express luciferase? Or is this a control?

Supplemental Figure 2C Are these cells primed with LPS?

Supplemental Figure 2D Molecular weight marker is missing

Supplemental Figure 2E Again are the white bars LPS primed? An unstimulated control i.e. doxycycline treated but with no LPS should be included here.

Line 151 Which panel of Supplemental figure 3 is being referred to?

Supplemental Figure S3A – Why is the 1-572 mutant being used here? This isn't referred to in the text. I have to assume that it's an NLRP3 truncation that is supposed to mimic the NLRP4 truncation referred to in line 147-149?

Supplemental Figure S3B – Blots are not labelled and are missing molecular weight markers.

Line 162 What is the rationale for making these particular truncations (620-720)? How were they chosen?

Figure 2 – Why are R258W and T346M mutants being used? I know that they are CAPS associated mutations but this has not been described anywhere in the text.

Figure 2C – What I assume to be the blot for NLRP3 is very overexposed. A lower exposure that gives a more representative reflection of expression levels should be presented.

Figure 3B These blots need molecular weight markers. Why is there less pro-IL-1 β in ATP and nigericin treated samples in empty vector and 1-667 when none of the pro-IL-1 β has been processed?

Figure 4B Are these NLRP3-YFP tagged proteins functionally the same as un-tagged version? Can they trigger IL-1 β production/death? Could NLRP3 localisation also be tested using an antibody to (untagged) NLRP3?

Supplementary Figure S5A – The 1-688 variant induces around three times the amount of IL-1 β relative to full-length NLRP3 in this assay which is inconsistent with previous observations. Is this due to the N and C terminal tags?

Supplementary Figure S5B – The blot is unlabelled and has no molecular weight markers.

Supplementary Figure S5C and D – these graphs are so small they are not legible.

Figure 5B – Given that this result is quite surprising (and contrary to the published literature on NLRP3-NEK7 interactions), it needs to be confirmed in an endogenous system. The interaction may be an artefact of overexpression.

Figure 5 C and D – Are these cells stimulated with LPS? This isn't indicated in the figure or the legend.

Figure 5 and Supplementary Figure 5

The BRET data suggest that the residues between 667-688 are essential for NLRP3 oligomerisation. Could this be confirmed by another assay such as SDD-AGE or Native-PAGE?

Figure 6A

The idea here was to do a dose titration but 5-6/7 doses are already maximal. This should be

repeated, changing the dose escalation so that it actually covers multiple submaximal doses.

Line 352 refers to pharmacological targeting of NLRP3, but I don't think this was done in the current study.

COMMENTS OF AND RESPONSES TO REVIEWER 1

R1-general: Nod-like receptor (NLR) family, pyrin domain containing 3 (NLRP3) is involved in the early stages of inflammatory responses where it senses cellular damage or distress due to bacterial infection. In the manuscript, the authors have constructed more than 20 truncation variants of murine NLRP3, established their stable expression system in NLRP3 KO BMDMs (Tet-on system), and they identified that leucine-rich repeat (LRR) is not essential for the activation of NLRP3 inflammasome. Regarding their conclusion, the authors also have tested a LRR deletion mutant (called mini-NLRP3) act as the minimal responsive NLRP3 in the experimental conditions. Overall of my impression is that demonstrating absence or dispensable is a daunting task. As they mentioned in the introduction section (NLRP3 inflammasome is activated by a wide range of DAMPs), their experimental conditions might be a limited range of monitoring (mostly using nigericin and provided data were preliminary). And most importantly, a functional role and /or physiologic relevance of LRR were not identified/demonstrated in the manuscript. For example, how the mini-NLRP3 sense DAMPs without LRR (molecular mechanism)?

Authors' response: We would like to thank the reviewer for comments. A variety of diverse triggers (soluble: nigericin, ATP, imiquimod; particulate: silica, alum) have been used in this study and in all those cases the results were the same, therefore we drew conclusion that LRRs in NLRP3 are dispensable for activation. In contrast to LRR domains of TLRs there is no evidence that LRRs of NLRP3 are involved in direct recognition of the triggers. Currently, the majority of NLRP3 canonical triggers cause K^+ depletion, and upon K^+ depletion NLRP3 is activated. This type of activation resembles effector-triggered immunity mostly found in plants. One of the major points of this study is that LRR is dispensable for canonical NLRP3 activation, which we think is an important conclusion which will have an important impact on the current understanding of the mechanism of NLRP3 activation that has been based primarily on previous overexpression studies. Our results modify the widespread perception on the role of LRRs as sensor or domain involved in autoinhibition and will be therefore interesting for a wide audience of immunologists.

We cannot rule out the possibility that the LRR domain of NLRP3 is important for some other, likely less emphasized functions, such as e.g. in the role of NLRP3 as a transcription factor, localization, stability, cell-specific role or alternatively that it may be an evolutionary remnant from its molecular ancestors. Since it is not known which is the direct molecular species that may be common to diverse NLRP3 agonists determination of its binding site is beyond the scope of this study. Nevertheless we can with confidence map it to the miniNLRP3.

These are specific comments for the authors which must be addressed.

R1-1). BRET experiments: What does the spike (decreasing BRET signal after adding nigericin) means? If the nigericin sensing site in NLRP3 have already identified based on

crystal structure, it would be great to make a non-sensing NLRP3 mutant and test the BRET experiment.

Authors' response: BRET has been used previously for monitoring NLRP3 conformation change or oligomerization (cited in the manuscript). Upon nigericin addition, the distance between the acceptor and donor is changed, indicating conformational change. As written above, there is currently no evidence for direct binding of known activators to NLRP3.

R1-2). Immunofluorescence images: In Figure 3C, the represented images were less resolution. And also, some positive controls (WT BMDMs?) must be provided.

Authors' response: To detect small speck formation, Z-stacks are recorded by the microscope and maximal intensity of the layers in the stack is used. The concentration of phalloidin stain close to membranes, however, results in blurring of the image. Images were now processed in Image J and the bottom and top layers in the phalloidin channel were excluded from the image, which resolved the problem. All images were processed in the same way. In this experimental setup, the full-length hNLRP3 is used as a positive control. Nevertheless, data on wild-type immortalized BMDMs are now provided in Suppl. Fig. 10.

R1-3). NLRP3 mutants (Cys mutants, T346M, R258W): It was unclear for me that how those mutants fit into the story of the activation of NLRP3 inflammasomes.

Authors' response: Pathological variants e.g. T346M, R258W are known to be constitutively active and were in the first place used to demonstrate the response of a constitutively active variant in comparison to deletion variants and whether they differ with respect to the role of the LRRs. As these variants do not need a specific trigger they were subsequently used to test the model of NLRP3 activation (Fig. 8).

There are many publications linking oxidative stress to the NLRP3 activation, thus Cys residues in the region 665-686 were mutated to probe whether any of those cysteine residues may be crucial for the NLRP3 activation and if dysfunctionality of 1-665 is based on the absence of those amino acids.

In the revised version of the manuscript the rationale for the selection of those mutants and experiments is better explained to improve the clarity and the flow of the report.

R1-4). Physiologic relevance of LRR in NLRP3: It is interesting to know that a phenotypic role of natural truncated version of NLRP3. To test this question, your system (knock in mice; mini-NLRP3) would be a great advantage.

Authors' response: The functionality of miniNLRP3 in vivo has been tested in a mouse peritonitis model (Fig. 4d). Cytokine levels in the peritoneal lavage are now provided in the Suppl. Fig. 11. Macrophages harboring miniNLRP3 and the full-length NLRP3 when injected i.p. induced higher IL-1b levels (Suppl. Fig. 11) and infiltration of neutrophils (Fig. 4d) in comparison to a signaling deficient variant or empty vector transduced cells,

confirming results in the cell culture which clearly demonstrate that in the absence of LRRs the NLRP3 is fully functional.

R1-5). Supplementary figure: Full blot images (western blots) in the manuscript were missing.

Authors' response: All full blot images were added to Supplementary information

COMMENTS OF AND RESPONSES TO REVIEWER 2

Reviewer 2 general: In this study Hafner-Bratkovic and colleagues address potential mechanism of NLRP3 inflammasome activation and oligomerization by employing NLRP3 truncation and chimeric variants in reconstituted macrophages to mimic endogenous levels. The authors show that the C-terminal LRR domain of NLRP3 is dispensable for activation and self-regulation, while the N-terminal pyrin domain is required to maintain NLRP3 in the inactive conformation. The authors also demonstrate that unlike NLRC4, active NLRP3 does not engage wildtype molecules to oligomerize and such lack of signal amplification was reasoned to be a protective mechanism to inhibit auto-inflammatory conditions.

The studies presented here are well designed and potentially provide novel insights into NLRP3 function that are significantly different from the current paradigm. However the approach of assessing truncation mutations has limitations and some type of complementary approach assessing endogenous proteins would improve the study.

Authors' response: First of all, we would like to thank the reviewer for his suggestions. Since the truncated variants maintain the full functionality with respect to activation by several agonists there can be no other explanation apart from the conclusion that NLRP3 retains virtually full functionality also in the absence of LRRs. BRET experiments on the conformational changes and examination of stability of variants demonstrated similar properties as for the full length NLRP3. Currently there is little information on the detailed molecular mechanism and structure of the NLRP3 inflammasome assembly. One of the major bottlenecks is that the introduction of NLRP3 inflammasome components into easily manipulated cells such as HEK293 easily leads to the constitutive inflammasome assembly. Our NLRP3^{-/-} iBMDM-based system mimics endogenous systems well, which we demonstrated on several occasions. Although our truncated variants were designed to provide the mechanistic insight, literature indicates that similar variants in fact occur in nature either through stop codon mutation or through alternative splicing, which we now discuss in the manuscript.

R2-1) In Figure 1 the authors reason loss of NLRP3 activity to low protein expression/degradation of truncation variants 1-850 to 1-996. Although diminished protein is observed in Figure 1D there is still substantial expression of these truncation variants. Hence it does not make sense that no IL-1beta secretion is measured from the truncation variants 1-850 to 1-996. The authors need to perform additional studies to

define the reason for this defect. These data also raise the possibility that LRR domains confer stability to NLRP3.

Authors' response: Additional studies were performed. First, to quantify protein levels, FL, and two variants with low expression levels were expressed by increasing amounts of doxycycline. All protein levels were normalized to FL protein level induced by 1 ug/ml doxycycline. This experiment showed that the level of expression of both mutated variants is below the level of NLRP3 protein needed to produce IL1b release (Suppl. Fig. 4) Further we demonstrated that those particular variants have lower protein life-time. Upon removal of doxycycline, the amount of those protein variants decreased, and the effect was even more pronounced when protein synthesis was blocked. In comparison, the full length NLRP3 remained stable even in the presence of protein synthesis inhibitor for 6h (Suppl. Fig. 5a). Further we show that the proteasome inhibitors partially reconstitute levels of mutants, demonstrating that those truncations are rapidly degraded by the proteasome system (Suppl. Fig. 5b). Since the amount of the produced miniNLRP3 is comparable to the amount of wtNLRP3 and higher than in case of those particular truncation mutants, the LRRs by themselves do not seem to be required for the protein stability and the decreased stability of those particular mutants is likely due to the exposure of hydrophobic patches or other signals of LRR truncation mutants that lead to the ubiquitination and degradation.

R2-2) In Figure 1B a number of the truncation mutants induce ~700-800 pg/ml of IL-1beta with LPS priming suggesting that they may be driving spontaneous activation of the NLRP3 inflammasome. In addition the length of priming (11 hours) is quite long. Have the authors tried these experiments with a shorter priming time (4-6 hours)?

Authors' response: The mentioned low (less than 10-20% of WT NLRP3-induced) amount of IL1b that has been observed in a couple of experiments could be unrelated to NLRP3 as it is sometimes observed even with empty vector cells. The amount of IL-1b release caused by the constitutively active variants is much higher (Fig. 2).

Priming duration depends on the the concentration of LPS, on the presence/absence of FBS, and also on the purity and type of LPS. Studies which use short priming times usually use very high amounts of LPS in an FBS-supplemented medium. We tested the activation of several NLRP3 variants depending on the priming time (Suppl. Fig. 8a), which showed maximal response for both FL and miniNLRP3 in interval from 10-15 h when LPS and doxycycline are used simultaneously. Additionally, as suggested by the reviewer, variants were initially treated by doxycycline for several hours, after which the LPS was added for shorter priming time with the same results regarding miniNLRP3 and FL activation as in the initial experiments (Suppl. Fig. 8b). Additionally, we show that miniNLRP3 and FL respond to nigericin even if the priming is done with P2CSK4 (TLR2/6 agonist) (Suppl. fig. 8c).

R2-3) Throughout the manuscript it would be helpful to include control cytokines such as

TNF and IL6 to show the affect of the truncation mutations are specifically on NLRP3 inflammasome and IL-1beta.

Authors' response: Additional cytokine data is now provided as suggested (Fig 3c, Suppl. Figs. 3, 7).

R2-4) In Figure 4A it would be helpful to provide additional supportive data such as the levels of proinflammatory cytokines in the peritoneal lavage including IL-1beta levels. Also in the experimental condition used where is the NLRP3 activating signal coming from?

Authors' response: After 24h IL-1 β was not detected in the peritoneal lavage (not shown), thus the experiment for cytokine detection was performed 2h post injection and the results on proinflammatory cytokines (IL-1 β , TNF- α , IL-6) have been added (Suppl. Fig. 11). Comparable increase in TNF- α and IL-6 levels was observed for all variants, which is expected as those cytokines are secreted in response to the LPS priming. The level of IL-1 β is increased in the peritoneal lavage of mice treated with cells harboring miniNLRP3 and the full-length NLRP3. The NLRP3 activating signal comes from the detachment of primed macrophages. There are several reports on the involvement of cell volume regulation (Compan et al., 2012) and ion channels in NLRP3 activation (several members of TRP family and others), which is likely the physiological trigger in this case. A more detailed description of the animal model has been included in the Methods and Results sections.

R2-5) In Figure 4B data the scale bars in each panel are of varying size - all panels should be the same magnification.

Authors' response: This has been corrected as suggested.

R2-6) Figure 5 - the authors should consider performing co-ip panel with 1-665, 1-688, and FL NLRP3 variants assessing other inflammasome components (ASC, caspase-1) involved in complex formation in addition to NEK7.

Authors' response: ASC immunoprecipitation is a difficult task as ASC forms aggregates rapidly after the inflammasome initiation. We failed to immunoprecipitate ASC when using anti-NLRP3 antibody Cryo-2, which binds to the PYD domain. However, we were able to observe ASC bands after immunoprecipitation of nigericin-treated inflammasome sufficient variants (miniNLRP3 and FL) when using anti-GFP antibodies which pulled down C-terminally tagged NLRP3 variants (Fig. 4c).

R2-7) The axis label for Figure 5E has been cut off.

Authors' response: We apologize for this, which has now been corrected.

COMMENTS OF AND RESPONSES TO REVIEWER 3

R3-general) The authors present a study on the molecular mechanisms of NLRP3 activation. This is an extremely exciting and much debated area in the field of inflammasomes and inflammation. A key finding is that the LRR domain of NLRP3 does not appear to function as a sensor domain or auto-inhibitory domain, as a truncation mutant lacking the entire LRR domain is able to reconstitute NLRP3-deficient cells with the same phenotype as full-length NLRP3. This is an interesting and novel observation, which will influence thinking in the field. Further truncations are performed and the minimal length of NLRP3 that retains function is found to be NLRP3 1-686 or 'mini-NLRP3'. This 'mini-NLRP3' is defined as 1-688 or human NLRP3 (when expressed in mouse cells). A number of cysteine residues in NLRP3, that may be expected to be involved in redox regulation, were mutated but did not result in any decrease in NLRP3-dependent IL-1 β secretion in this system. Surprisingly, 'mini-NLRP3' was able to interact with NEK7 in an overexpression system. Experiments using intramolecular and intermolecular BRET found that 'Mini-NLRP3' was also able to interact with itself and full-length NLRP3 when triggered by nigericin. Finally, the authors propose that the PYD domain of NLRP3 is inhibitory and maintains NLRP3 in an inactive conformation as replacing NLRP3 PYD with a CARD domain renders the protein hyperactive in a manner similar to CAPS mutations. The authors extensively exploit the doxycycline inducible retroviral expression system that they have developed to make some interesting and challenging observations on the way NLRP3 is activated.

Overall assessment: The manuscript has some interesting observations about NLRP3 signalling/function, but the manuscript as it currently stands really lacks clarity and flow. There are general issues with language and continuity, the rationale for particular experiments is missing, and the manuscript figures are extremely difficult to follow (many are missing labels, legends etc so the reader often has to guess what the data is). Many of the manuscripts figures require additional experiments to justify the author's claims. This manuscript would only be suitable for publication in Nat Comms if the authors were willing and able to perform really substantial revisions to bring the manuscript up to the required quality.

Authors' response: We are grateful to reviewer for his extensive review and for his valuable comments upon which the manuscript has been substantially rewritten to improve the clarity and flow and to explain the intentions and results in more detail. The revised manuscript was also checked for grammatical errors by a professional text editing company. A variety of additional experiments suggested by the reviewer and also by us were performed that further strengthen the conclusions. The introduced modifications are described in more detail below.

Major comments (MC):

R3-MC1: This study makes extensive use of truncation mutants to map domains required for NLRP3 activation and function. Such experiments, while extremely useful, need to be

performed and interpreted with caution – mutation can have small effects on NLRP3 protein expression that can have a large effect on NLRP3 function (as expression level is critical for NLRP3 signalling), and mutations may have unintended consequences on gross protein structure. While the former (expression level) is somewhat addressed, the latter is not addressed at all. Can the authors be sure that non-functional NLRP3 truncation mutants are folded correctly (particularly in the NACHT domain that is critical for signalling)?

Authors' response: Both aspects of possible mutation effects mentioned (expression level and the folding of NACHT domain) were additionally experimentally addressed. In summary, we demonstrate that the truncation of a couple of the C-terminal LRRs leads to rapid NLRP3 variant clearance by proteasome system, which results in expression level of NLRP3 variant insufficient for activation. Further we show that 1-665 is able to bind to ATP beads, which can be outcompeted by CY-09, demonstrating that the NACHT domain of 1-665 is correctly folded and functional as far as ATP binding is concerned. As mentioned in the response to reviewer#1, "since several truncated variants maintain the full functionality with respect to activation by several agonists there can be no other explanation apart from the conclusion that NLRP3 is functional also in the absence of LRRs".

R3-MC2: Figure 1: The retroviral expression system seems to be appropriate for studying the role of the LRR domain in the regulation of NLRP3. While the results clearly show that truncation of the whole LRR domain (731) does not result in loss of function, the results with the truncations from 850-996 are puzzling, and cannot be simple explained away by a relatively small decrease in the level of protein expression. These expression levels would need to be quantified (relative to their loading controls) perhaps in comparison to the level of WT FL NLRP3 expression induced by different levels of doxycycline as presented in Supplemental Figure 2 C and D. What is the minimal level of expression required for NLRP3 to function in this system? Are these truncations (850-996) expressed below this level? Could one of the truncations (850-996) be expressed at a higher level to demonstrate that it is still functional? This is a fundamental observation in the study and these issues need to be addressed.

Authors' response: As written in response to reviewer 2, to quantify protein levels, FL, and two variants with low expression levels were expressed by increasing amounts of doxycycline. All protein levels were normalized to FL protein level induced by 1 ug/ml doxycycline. This experiment showed that the level of expression of both mutated variants is below the level of NLRP3 needed to produce IL1b release (Suppl. Fig. 4). The concentrations of doxycycline above 10 µg/ml have not been used as they interfere with priming and cell well-being. Further we demonstrated that variants have lower protein life-time. Upon removal of doxycycline, the amount of mutant proteins decreased, and the effect was even more pronounced when protein synthesis was blocked. In contrast, the full length NLRP3 level remains stable even in the presence of a protein synthesis inhibitor for 6h (Suppl. Fig. 5a). Further we show that proteasome inhibitors partially increase

the amount of mutant proteins, demonstrating that those truncations are rapidly degraded by the proteasome system, most likely due the exposure of hydrophobic patches that result from the LRR truncations. As the mini-NLRP3 with deletion of all LRRs is functional we reason that the decreased stability and functionality of those particular truncated variants are not really relevant for the understanding of the mechanism of NLRP3 inflammasome activation. In fact we think that the manuscript would not lose any of its “take home message” by omitting the results with shorter LRR deletions, nevertheless we kept them due to the completeness of the study.

R3-MC2: Line 176 and Supplemental Figure S4: As the expression of all the NLRP3 variants is under the control of doxycycline there is no relevant transcriptional regulation of NLRP3 as these are NLRP3-deficient cells. Post-transcriptional priming of NLRP3 is a rapid event (within 5 mins of LPS treatment) so I don't see any relevance to looking at long LPS priming times. To address whether ‘mini-NLRP3’ is post-transcriptionally regulated in a similar manner to full length NLRP3 an experiment should be performed using short priming times with LPS (e.g. 0-30 min) and nigericin stimulation. Cell death and/or caspase-1 activation can then be measured as a readout of NLRP3 activation.

Authors' response: Longer priming times are required for the transcription/translation of proIL1 β , if we want to use IL1 β as a readout. In most cases, within the study we use doxycycline and LPS simultaneously, so during this period also NLRP3 expression is induced. We also observed that at lower doxycycline concentration the LPS priming enhanced the protein level, but this effect is not specific to NLRP3 as it has been observed with F-luciferase as well. Optimal time of priming when using IL-1b as a readout has been estimated between 10-15h under the conditions using 100ng/ml Ultrapure LPS and doxycycline (Suppl.Fig. 8a). In the revised version of the manuscript we also provide data with shorter priming times (Suppl. Fig. 8b), when NLRP3 expression was preinduced. Additionally, we demonstrate that another priming agent (P2CSK4) could successfully replace LPS (Suppl.Fig. 8c). We also performed the experiment suggested by the reviewer, however, no NLRP3-dependent cell death (monitored by LDH release) has been observed with very short LPS priming times (15 and 30 min) (data not shown).

R3-MC3: As ‘mini-NLRP3’ lacks numerous sites for post-translational modification that have been described in the literature (e.g. PTPN22, PKA, BRCC3) this is something that needs to be addressed.

Authors' response: As mentioned by the reviewer, NLRP3 activation seems to be tightly regulated by posttranslational modifications, many of the sites are present both in mini-NLRP3 and the FL (such as for PKA, BTK...), while in some cases also LRR domain is targeted, such in the case of BRCC3 and PTPN2.

To probe for this role of the LRRs, BRCC3 selective inhibitor G5 was used, which inhibits NLRP3 activation of both mini NLRP3 and wtNLRP3, which is in agreement with the study which showed that BRCC3 targeted both LRR and NACHT domains (Py et al., Mol

Cel, 2012). To address the action of PTPN22, which targets the 861 site, which is absent in the miniNLRP3, LTV-1 inhibitor has been used. LTV-1 inhibitor only inhibited both NLRP3 variants at high concentrations (not shown). Unfortunately, LTV-1 is a selective inhibitor of human PTPN2, while it inhibits mouse PTPN2 only at higher concentrations, where also other kinases are inhibited. Although miniNLRP3 lacks some sites for the posttranslational modification no significant differences in activation were observed in the cell culture. Results and discussion on the potential role of the posttranscriptional modifications of the NLRP3 is included in the manuscript.

R3-MC4: Figure 2: Can you comment on what domains/features are present in NLRP3 686 (minimally responsive) and NLRP3 665 and NLRP3 572 (Supp fig S3A) that fail to respond?

Authors' response: Based on the sequence alignment, 686 and 665 variants should consist of the PYD-NBD-HD1-WHD-HD2 domains, while 1-572 consists of PYD-NBD-HD1-WHD. The molecular model of NLRP3 predicts a short helical element in the region 665-686 (which is currently in the absence of known structure of NLRP3 just a speculation). So far point mutations within 665-686 have not identified a defined functional site (Fig. 5B) responsible for the difference.

R3-MC5: Figure 3C Although it is clear that NLRP3 1-667 does not induce ASC speck formation in response to LPS/Nigericin it is hard to determine whether the levels of ASC expression are similar in these different cell lines. This should be addressed with western blot or quantification of microscopy data. Some of the panels don't appear to be in focus (eg. 1-688 LPS). Although NLRP3 interacts with ASC via it's PYD domain it would be interesting to examine whether the NLRP3 truncations presented here interact with ASC to the same extent using for example a co-immunoprecipitation assay.

Authors' response: The experiment was repeated to include ASC expression (Fig. 3b) and, as mentioned in response to reviewer #1, microscopy panels were processed to exclude layers that contributed to blurring. ASC immunoprecipitation is a very difficult task as it forms aggregates rapidly after inflammasome initiation. We failed to immunoprecipitate ASC using anti-NLRP3 antibody Cryo-2, which binds to the PYD domain. However we were able to observe ASC bands after immunoprecipitation of nigericin-treated inflammasome-sufficient variants using anti-GFP antibodies which pulled down C-terminally YFP-tagged NLRP3 variants (Fig. 4c). For an estimation of ASC recruitment we provide the analysis of ASC speck formation (Fig. 3e), which is a widely accepted functional assay and corresponds to other inflammasome-related readouts.

R3-MC6: Figure 4A I am unfamiliar with this model of peritonitis – could some reference be provided for previous examples of its use? What statistical test was performed?

Authors' response: To our knowledge this is the first use of cell implantation method for inflammasome studies, thus the animal model is described in more detail in Methods and

Results section. Cytokine levels in the peritoneal lavage are reported in Suppl. fig. 11. Unpaired two-tailed t-test with or without Welch correction (for comparison of data with different variances) was used for pairwise comparison, as stated in the Methods section.

R3-MC7: Figure 6A: As the CARD domain of NLRC4 and ASC can interact directly with caspase-1 it is possible that expression of these chimeric proteins can trigger caspase-1 autocatalytic processing and activation. The formation of ASC specks in these cells needs to be assessed. An important control here would be to express NLRP3 with no PYD domain (rather than with the PYD domain replaced with a CARD).

It would be very interesting to truncate NLRP3 from the N-terminus to assess which elements of the PYD domain are required to maintain the inhibited state of NLRP3.

Authors' response: As suggested, dPYD control was included. PYD or CARD only controls were used as in overexpression experiments these domains tend to oligomerize on their own, thus constitutive activation could be due to the oligomerization of those domains, which is not the case. N-terminal truncations of NLRP3 have not been tested, because the loss of parts of PYD domain leads to the loss of ASC binding, hence no activation could be observed. There are several studies reporting on the importance of the N-terminal residues of NLRP3 for the stabilization of PYD domain (Bae&Park, 2011), interaction with ASC (Lu et al., Cell, 2014) and interaction with MAVS (Subramanian et al., 2013).

To determine whether the chimeric proteins engage ASC, ASC specks were detected. Nevertheless, even NLRC4 activation leads to ASC speck formation, yet it is not absolutely needed for the caspase-1 processing. To unambiguously answer the question whether chimeras are able to activate caspase-1 in the absence of ASC, a novel ASC-KO-Tet3G single clone cell line was produced and chimeric proteins were introduced. Our results show that chimeras are to some extent also able to drive IL-1 β production in the absence of ASC, which means that they are able to some extent to engage caspase-1 directly. All this data can be found in New Fig. 7 and Suppl. Fig. 14).

R3-MC8: Figure 6B This is an intriguing experiment and although it hints at the stoichiometric model of NLRP3 oligomerisation there are a number of caveats. It is not really definitively known that the point mutations used are in an 'open conformation'. This data needs to be accompanied by data showing the expression level of NLRP3 in each case, or the results cannot be interpreted.

Authors' response: The term open conformation has been changed to a more appropriate 'activated form'. As suggested, this data is now accompanied by an analysis of protein expression (Fig. 8), which demonstrates that expression of pathological variants is indeed low compared to endogenous NLRP3 upon LPS priming.

R3-MC9: The ATPase activity of NLRP3 is known to be essential to its function. Is it possible that these mutations have affected the nucleotide binding/hydrolysis ability of NLRP3?

Authors' response: We are grateful to the reviewer for pointing this out. Based on his suggestion we performed additional experiments to determine the ATP binding propensity of truncation mutants (Fig. 5c), which additionally strengthens our conclusions. Nucleotide binding functionality has been tested by an ATP-bead-pulldown assay. 1-665, 1-686 and the full-length NLRP3 are capable of binding ATP, which can be outcompeted by CY-09, an NLRP3 selective inhibitor, which binds to the ATP-binding site.

Minor Comments:

Line 1-2 The title is extremely unclear and should be changed to something that more clearly indicates the experimental results.

*Authors' response: The title has been changed to '**NLRP3 lacking leucine-rich repeat domain can be fully activated by the canonical inflammasome pathway**'.*

Line 28 'self-encoded' is not correct (e.g. ATP is not an 'encoded' molecule), 'self-derived' is more appropriate.

Authors' response: Corrected as suggested.

Line 29 and Line 32 The Leucine-rich... the use of the definite article throughout the manuscript needs to be checked.

Authors' response: The manuscript has now been checked for grammar, article use and typos by a professional editing company.

Line 43-45 The list of abbreviations is incomplete. Some that are missing include: DAMPs, PAMPs, BIR, LRR.

Authors' response: The list of abbreviations has been extended.

Line 73 NEK7 is not regarded as an NLRP3 'ligand' – it's a protein that interacts with NLRP3 and is required for NLRP3 function.

Authors' response: This sentence has been rewritten to ''NEK7 kinase interaction with NLRP3 was shown to be necessary for NLRP3 inflammasome activation''

Line 78 This phrasing implies that HET-E and TP-1 are NLR proteins which they aren't.

Authors' response: The definition of NACHT has now been added to abbreviations section and text rewritten to avoid such misunderstandings.

Line 96 What does an ‘inflammasome particle’ refer to? Please define this or be more specific.

Authors’ response: This sentence has been rewritten to ‘leading to the assembly of substoichiometric oligomers’.

Line 110 ‘LRR-deficient minimal NLRP3’ is quite a confusing term. Perhaps something like NLRP3 without the LRR domain would be clearer?

Authors’ response: This sentence has been rewritten for the sake of clarity.

Line 125 what does ‘available data’ refer to? Is this published work?

Authors’ response: The reviewer is correct. ‘Available data’ has been changed to ‘published work’.

Figure S1 Needs much clearer labelling. For example, in the NCBI schematic what is the F beside NBD? In the Albrecht model what are 1-2-3 to the right of the NBD? In the Proell model what do C-WH and SH2 refer to? These are not defined anywhere.

Authors’ response: Explanation/labeling is now provided in S1 and figure legend.

Figure 1 A What does AD mean? Perhaps this figure would be clearer if the LRR domains were first defined in a consistent manner and then numbered 1-9.

Authors’ response: AD means associated domain, which has been written in full in the figure. LRR were labeled 1-9 as suggested in Fig.1 and 2.

Figure 1 B-C For only two biological replicates – presenting the mean +/- range rather than standard deviation may be more appropriate.

In these bar charts and throughout the rest of the figures it is never indicated what the different coloured bars refer to – I can only assume that the white bars here are LPS primed cells? For a reader to be able to interpret this data it really needs to be labelled.

Authors’ response: All data points are now shown (as required by the Nat. Commun.). In revised version we relabeled bars so that white bars are LPS-treated throughout the manuscript. Additionally, we agree that a legend in the figure (not just in figure legends) makes the manuscript more easily interpreted, thus it has been inserted as well.

Figure 1D Label the blot for NLRP3.

Authors’ response: Done as suggested.

Supplemental Figure 2A – NLRP3 deficient cells are conventionally termed Nlrp3^{-/-} and this should be used in the manuscript.

Authors’ response: NLRP3-deficient or NLRP^{0/0} has been changed to NLRP3^{-/-} throughout the manuscript

Supplemental Figure 2B – Why is luciferase activity being measured here? Does the NLRP3 expression construct also express luciferase? Or is this a control?

Authors' response: This is just a control luciferase. Figure legend for Suppl. Fig. 2 has been rewritten to clarify this.

Supplemental Figure 2C Are these cells primed with LPS?

Authors' response: Cells were LPS-primed, which is now addressed in the Figure legend and in the figure itself.

Supplemental Figure 2D Molecular weight marker is missing

Authors' response: Molecular weight marker is indicated.

Supplemental Figure 2E Again are the white bars LPS primed? An unstimulated control i.e. doxycycline treated but with no LPS should be included here.

Authors' response: Cells were LPS-primed, which is now addressed in the Figure legend and also in the figure itself. The control (dox-treated, no LPS) is now included in the figure (Suppl. Fig. 2c).

Line 151 Which panel of Supplemental figure 3 is being referred to?

Authors' response: To panel B, in revised manuscript this is Suppl. Fig. 6B, which has been indicated in the main text.

Supplemental Figure S3A – Why is the 1-572 mutant being used here? This isn't referred to in the text. I have to assume that it's an NLRP3 truncation that is supposed to mimic the NLRC4 truncation referred to in line 147-149?

Authors' response: According to published studies this variant should contain PYD-NBD-HD1-WHD domain and indeed refers to constitutively active variant of NLRC4. Note that variants of similar predicted composition were used as NLRP3 devoid of LRR. This has now been included in the manuscript. Due to common usage of NACHT domain as synonym for NBD, we now adhere to nomenclature used by structural studies on NLRC4 and NLRC2, where NBD is used and NBD-associated domains are named HD1, WHD and HD2.

Supplemental Figure S3B – Blots are not labelled and are missing molecular weight markers.

Authors' response: Blots were labeled and molecular weight marker inserted (Suppl. Fig. 6b).

Line 162 What is the rationale for making these particular truncations (620-720)? How were they chosen?

Authors' response: Those variants were chosen based on the model of NLRP3 based on NLRC4 and to include potentially interesting motifs (this is now explained in the Results section).

Figure 2 – Why are R258W and T346M mutants being used? I know that they are CAPS associated mutations but this has not been described anywhere in the text.

Authors' response: They are used as a control for constitutive activation, which is now explained in Results section.

Figure 2C – What I assume to be the blot for NLRP3 is very overexposed. A lower exposure that gives a more representative reflection of expression levels should be presented.

Authors' response: A blot with shorter exposure is now shown.

Figure 3B These blots need molecular weight markers. Why is there less pro-IL-1 β in ATP and nigericin treated samples in empty vector and 1-667 when none of the pro-IL-1 β has been processed?

Authors' response: Molecular weight markers were added. Although an interesting question, the effect is clearly not dependent on the NLRP3. The decrease of pro-IL-1b observed is lower than in inflammasome competent cells (reconstituted with 1-688 and FL). Also in current blot (experiment was repeated to provide also ASC labelling) this effect is not observed. This Western blot clearly indicated that LPS induced levels of pro-IL1b are comparable among different cell lines, so the effect of absent release of IL-1b in 1-667 does not come from low pro-IL-1b expression.

Figure 4B Are these NLRP3-YFP tagged proteins functionally the same as un-tagged version? Can they trigger IL-1 β production/death? Could NLRP3 localisation also be tested using an antibody to (untagged) NLRP3?

Authors' response: Yes, they are. This data is now provided as Fig. 4a. Anti-NLRP3 antibody that we use binds to PYD domain, which interacts with ASC. We tested immunolabelling prior to preparation of tagged variants, but had difficulty discriminating background staining from the real signal.

Supplementary Figure S5A – The 1-688 variant induces around three times the amount of IL-1 β relative to full-length NLRP3 in this assay which is inconsistent with previous observations. Is this due to the N and C terminal tags?

Authors' response: Introduction of large tags at the N-terminus could in principle interfere with inflammasome assembly, however, in this case lower IL-1b activation by doubly tagged WT NLRP3 is most likely due to lower protein expression (Suppl. Fig. 12B).

Supplementary Figure S5B – The blot is unlabelled and has no molecular weight markers.

Authors' response: Blot was labeled and molecular weight marker inserted

Supplementary Figure S5C and D – these graphs are so small they are not legible.

Authors' response: Figure S5 has been divided into two Figures to increase the panel size (Suppl. Fig. 12 and 13).

Figure 5B – Given that this result is quite surprising (and contrary to the published literature on NLRP3-NEK7 interactions), it needs to be confirmed in an endogenous system. The interaction may be an artefact of overexpression.

Authors' response: NEK7 was weakly pulled down from LPS-primed macrophages, but NLRP3 interaction with NEK7 upon nigericin treatment was enhanced only for inflammasome sufficient variants (Suppl. Fig. 9b). Note that a recent study by He et al., 2018 also suggests on the presence of NEK7 interaction N-terminally to LRR.

Figure 5 C and D – Are these cells stimulated with LPS? This isn't indicated in the figure or the legend.

Authors' response: Yes, cells were primed, which we now included into figure legend.

Figure 5 and Supplementary Figure 5

The BRET data suggest that the residues between 667-688 are essential for NLRP3 oligomerisation. Could this be confirmed by another assay such as SDD-AGE or Native-PAGE?

Authors' response: Native PAGE was performed. We observed substantial changes in the mobility pattern upon LPS priming, but upon nigericin treatment could not observe discrete bands that could be undoubtedly defined as NLRP3 oligomers, thus this experiment was not included in the manuscript.

Figure 6A

The idea here was to do a dose titration but 5-6/7 doses are already maximal. This should be repeated, changing the dose escalation so that it actually covers multiple submaximal doses.

Authors' response: Experiment was performed as suggested, please refer to Fig. 7a

Line 352 refers to pharmacological targeting of NLRP3, but I don't think this was done in the current study.

Authors' response: This refers to K⁺ efflux inhibition (i.e. using the drug glyburide) and ROS inhibition, figure 2f (Fig. 2g in revised manuscript). Additionally, we now add chart Fig. 2h, which shows inhibition by inflammasome inhibitors, MCC950, Cy-09 and shikonin.

Reviewers' comments:

Reviewer #1 (Remarks to the Author):

The authors have addressed my initial concerns about the functional role of LRR in NLRP3 via several control experiments and much strength their conclusions.

Reviewer #2 (Remarks to the Author):

The authors have addressed my previous concerns.

Reviewer #3 (Remarks to the Author):

The authors have made a significant effort to address all the reviewers' comments which is admirable. The manuscript is much improved by revision, but there remain significant issues that currently preclude publication in an excellent journal such as Nat Comms.

Major comments:

1. There are still some grammar issues and there are still issues throughout with labelling of figures, inconsistent call out of figures and missing references (see minor comments).
2. With reference to the lower expression of some of the LRR truncations in Figure 1 this is addressed in new Supp figs 4 and 5 where truncations 879 and 996 are compared to full length NLRP3. These truncations definitely don't express as well as FL despite increasing doxycycline concentrations (Supp Fig. 4). The truncations seem to be more sensitive to degradation in the presence of cycloheximide. In Supp fig 5b, levels of 879 and 965 do seem to decrease over time and this is partially rescued by proteasome inhibitors (more clearly for 879). Overall these unresponsive variants do seem to be getting degraded. However, it is still the case that the small changes in expression observed in Fig 1d have very large effects on function that may not be completely dependent on proteasome mediated degradation. This should be clearly acknowledged.
3. In Fig 2b and 2c the expression of truncations 665-620 is lower than that of 686 'mini-NLRP3'. In fact the expression of 665 is consistently lower than 686 (see Fig 4c, Fig 2c Fig 5c and Fig 6b). How can the authors decide that 665 is a non-functional truncation mutant, while other truncations (879 and 965) shown in Figure 1 and Supp fig 4 and 5 are assumed to be functional but quickly degraded? These interpretations are inconsistent with one another. The degradation/stability of 665 vs 686 needs to be addressed in order to properly interpret these data.
4. The NEK7 result is still not convincing. The endogenous IP shown in Supp Fig 9b is difficult to interpret as the levels of NLRP3 pulled down vary, and seem to correspond to the level of NEK7 that interacts. Could this perhaps be tried in the opposite manner – IP NEK7 and blot for NLRP3? Given this experiment is N=2 I think this should be repeated in a more controlled manner, as it contradicts previous publications.
5. Figure 4c – this is a strange result given that NLRP3 interacts with ASC via PYD-PYD domain interactions. The PYD domains of all of the constructs used are the same so why does 665 not interact with ASC?

6. Supp Figure 13 – this is surely a key observation that provides some rationale for why 667 (665) cannot signal. The expression levels of these constructs should be shown.

7. Figure 7c – specks should be quantified here.

Minor comments:

8. Some data is presented as the mean and SEM or SD of 2 biological replicates. Mean +/- range is more appropriate for n=2.

9. Lines 155-165: This text seems disjointed with data presented earlier in Fig. 1. The authors appear to have jumped forward to addressing other truncations shown in Supp fig 6 (none of which are used in Figure 1).

10. Lines 264-267 are referring back to Fig 3 but have been inserted after the discussion of Figure 4 (lines 255-263).

11. Supp Fig 14b and 14d are not referred to anywhere in the text.

12. Mini-NLRP3 (1-686 truncation) is referred to as mini-NLRP3 in the text but not in the figures. This is quite inconsistent for the reader.

13. Line 1-2: Suggest: NLRP3 lacking THE leucine-rich repeat domain can be activated by canonical NLRP3 inflammasome stimuli.

14. Line 56: what does 'in certain ways' refer to here? The comparison of TLR4 agonists and NLRP3 stimuli does not make sense.

15. Line 94: 'apoptosome and inflammasome assembly' this needs to be referenced.

16. Line 98: 'early reports' please reference these. 'the NLRP3 overexpression' delete 'the'.

17. Supp Fig 1: the legend is incorrect as for c and d it says: NLRPC4 (B) and NLRC2 (c)

18. Line 134 and Supp fig 3: There is clearly some variability in IL-6 and TNF α secretion with the various truncations (e.g. between 823 and 995 are lower than FL). These may be due to experimental variability, but unless statistics are performed on these data then it is inappropriate to say that 'none of the truncation variants affected the release levels'.

19. Line 144: 'While THE protein level'.

20. Line 145/Supp fig 5A: Where is the data showing removal of doxycycline results in decreased protein levels of 879 and 965? This blot would need to show levels prior to dox removal for comparison.

21. Line 159: Supp fig 6b is called out before 6a.

22. Line 173: 'the NLRC4' delete 'the'.

23. Line 178: 'published studies' should be referenced.

24. Figure 2e and 2f are too small and the x axis labels are not aligned – switch x axis labels to vertical.

25. Line 190 – same comment on Supp fig 7 as for Supp fig 3- there is some variation in TNF (Supp 7b) with the truncations.
26. Line 205 – the phenotype here is very similar but is not a ‘phenocopy’ or ‘the same phenotype’ Supp fig 8.
27. Line 242 - ‘upon the NLRP3’ delete ‘the’.
28. Figure 3e – what does %ASC speck/cell actually mean? Is this the percentage of the cells in the field of view that contain ASC specks? How many cells were analysed in total?
29. Figure 5c – the labelling here is not obvious – pulldowns with control or ATP beads need to be more clearly indicated.
30. Line 308 – ‘of the NLRP3’ delete ‘the’. Also delete ‘similarly’.
31. Supp Fig 12 d-f are called out before 12 c
32. Figure 6- Western blots (top panels) are not labelled.
33. Line 333 ‘for the inflammasome’, delete ‘the’.
34. Line 334 insert ‘the’ before ‘shorter truncation’ – ‘THE shorter truncation’.
35. Figure 7d – should be labelled as ASC-/- cells.
36. Figure 8 a-d – what cells are these? Please label these figures it’s impossible to tell what the difference is between a-d.
37. Line 417 – ‘the NLRP3’, delete ‘the’.
38. Line 425 – insert the ‘lacking THE LRR..’
39. Line 457 – insert the ‘THE present’

Reviewer #3 (Remarks to the Author):

The authors have made a significant effort to address all the reviewers' comments which is admirable. The manuscript is much improved by revision, but there remain significant issues that currently preclude publication in an excellent journal such as Nat Comms.

AUTHORS' RESPONSE: We would like to thank the reviewer for acknowledging the improvements and additional results of the first revision.

Major comments:

1. There are still some grammar issues and there are still issues throughout with labelling of figures, inconsistent call out of figures and missing references (see minor comments).

AUTHORS' RESPONSE: We are grateful to the reviewer for his/her effort in correcting grammatical issues (article use) and suggesting how the manuscript could be improved. In our defense, we need to say that manuscript has been edited for grammatical issues by a professional company twice (in the initial as well as for the revised version), so we believe that we have done everything what was in our power to present grammatically correct first revision. To avoid introduction of additional article misuses, we tried to make as little changes to the text as possible at this stage.

2. With reference to the lower expression of some of the LRR truncations in Figure 1 this is addressed in new Supp figs 4 and 5 where truncations 879 and 996 are compared to full length NLRP3. These truncations definitely don't express as well as FL despite increasing doxycycline concentrations (Supp Fig. 4). The truncations seem to be more sensitive to degradation in the presence of cycloheximide. In Supp fig 5b, levels of 879 and 965 do seem to decrease over time and this is partially rescued by proteasome inhibitors (more clearly for 879). Overall these unresponsive variants do seem to be getting degraded. However, it is still the case that the small changes in expression observed in Fig 1d have very large effects on function that may not be completely dependent on proteasome mediated degradation. This should be clearly acknowledged.

AUTHORS' RESPONSE: We agree with the reviewer that some window for other explanations should remain open, thus the sentence has been changed to:

"This effect, however, could be at least partially attributed to the lower protein levels of the truncation variants 1-850 to 1-996 (Fig. 1d)."

Additionally, a sentence has been included at the end of the paragraph:

"and instability of these NLRP3 variants could at least partially contribute to their unresponsiveness."

3. In Fig 2b and 2c the expression of truncations 665-620 is lower than that of 686 'mini-NLRP3'. In fact the expression of 665 is consistently lower than 686 (see Fig 4c, Fig 2c Fig 5c and Fig 6b). How can the authors decide that 665 is a non-functional truncation mutant, while other truncations (879 and 965) shown in Figure 1 and Supp fig 4 and 5 are assumed to be functional but quickly degraded? These interpretations are inconsistent with one another. The degradation/stability of 665 vs 686 needs to be addressed in order to properly interpret these data.

AUTHORS' RESPONSE:

We provide time course for protein expression in combination with proteasome inhibitors for FL, 1-686, and 1-665 variants in Suppl. Fig. 8e, showing similar observation with all three variants.

In addition, we believe that longer explanation is required to address this concern of the reviewer. First, the expression of truncated variants in 541-720 region should be compared not to 686 but to the expression of full length protein. It is true that in Figure 2c, the expression of 620-665 looks a bit weaker. This particular blot was included as it corresponds to the activation experiments in Figures 2b, 2e, and 2f. In other blots in the paper the expression of 665 is even higher than of full length variant (6a, 6b, 5c). In other blots the variant 665 (667) is expressed comparable or stronger than a variant 686 (688) (e.g. in Figures 3c, 4c) and stronger than FL.

Regarding dysfunctional truncations in the LRR we reasoned in the first revision that they are likely dysfunctional because they are degraded. According to the previous comment we state that degradation partially contributes to unresponsiveness. We cannot confidently state anything about the folding of those variants.

4. The NEK7 result is still not convincing. The endogenous IP shown in Supp Fig 9b is difficult to interpret as the levels of NLRP3 pulled down vary, and seem to correspond to the level of NEK7 that interacts. Could this perhaps be tried in the opposite manner – IP NEK7 and blot for NLRP3? Given this experiment is N=2 I think this should be repeated in a more controlled manner, as it contradicts previous publications.

AUTHORS' RESPONSE: We tried to pull-down NLRP3 variants with NEK7 antibodies before the submission of the first revision, but very high background (high control IgG pull-down) was observed. So to address this comment, we used YFP-tagged NLRP3 variants, which have also been used for the immunoprecipitation of ASC. In this case, we were able to pull-down NEK7 in complex with the inflammasome-competent variants (686 and FL) after the nigericin treatment (New Suppl. Fig. 9b). Both previous studies which reported mapping of the NEK7 binding site used HEK293 overexpression system (Shi et al., Nature immunology, 2015 mapped it to the C-terminus of LRR, while He et al, Nature, 2016 reasoned that both NACHT and LRR domains are necessary for NEK7 binding). As in HEK293 overexpression system even 665 variant pulls down NEK7, we believe it may not be the absence of the NEK7 binding site in 665 which prevents its immunoprecipitation upon nigericin treatment. A possible explanation could be that NEK7-NLRP3 interaction is stabilized in the inflammasome complex (Shi et al., Nat. Immunology, 2015 pulled down NEK7 with antibodies targeted to ASC in the absence of NLRP3) while for 665 variant it is only transient as the said variant does not form a functional inflammasome and we could thus not observe it.

Differences in the two first above mentioned studies, our results, and a recent study on oridonin, which covalently binds NLRP3 at Cys279 and inhibits NLRP3-NEK7 interaction (He et al., Nature Communications, 2018) demonstrate that a study specifically focusing on NEK7 binding site(s) on NLRP3 is needed, which however is not the scope of this study.

5. Figure 4c – this is a strange result given that NLRP3 interacts with ASC via PYD-PYD domain interactions. The PYD domains of all of the constructs used are the same so why does 665 not interact with ASC?

AUTHORS' RESPONSE: Previous studies (e.g. Tang et al, Nature Communications, 2017) demonstrated that ASC and NLRP3 are pulled-down together only after the addition of triggers. In NLRP3 inflammasome assembly, it is proposed that triggers induce the formation of proper NLRP3 oligomer(s), which nucleate ASC recruitment and polymerization (the models of inflammasome assembly are nicely described in Elliot & Sutterwalla, Immunol Rev, 2016). Our immunoprecipitation result demonstrates that in contrast to 665, the interaction of ASC with 686 or FL is strongly enhanced upon nigericin treatment, which is consistent with our results showing that nigericin treatment of cells harboring 665 (human 667) does not lead to ASC speck formation. It is thus possible that 1-665 fails to immunoprecipitate ASC as it is incompetent in proper oligomer formation that would successfully nucleate ASC recruitment and polymerization, and thus 665 fails to form inflammasome upon activation.

6. Supp Figure 13 – this is surely a key observation that provides some rationale for why 667 (665) cannot signal. The expression levels of these constructs should be shown.

AUTHORS' RESPONSE: Additional Western blot experiments have been performed and included as suggested. However, as Western blot has been performed on request of the reviewer (with novel isolations of plasmids etc.), also fluorescence and luminescence readings of that particular BRET experiment are shown to provide a correlation between the YFP-tagged and luciferase-tagged protein expression and activity.

For BRET experiments, transfections are done in larger formats and then seeded for fluorescence and luminescence readings. We should point out that BRET calculation is ratiometric and compensates for the differences in expression (luciferase activity). Thus, low BRET signal for 667 variants is not due to low 667 expression (which is comparable to 688 and for YFP-tagged variants higher than FL), but due to the failure of this variant to oligomerize.

7. Figure 7c – specks should be quantified here.

AUTHORS' RESPONSE: ASC speck quantification has been done as suggested and results are included in the Supplementary Figure 14 c.

Minor comments:

8. Some data is presented as the mean and SEM or SD of 2 biological replicates. Mean +/- range is more appropriate for n=2.

AR: In all cases all experimental data points are shown, which in the case of n=2 also represent +/- range borders.

9. Lines 155-165: This text seems disjointed with data presented earlier in Fig. 1. The authors appear to have jumped forward to addressing other truncations shown in Supp fig 6 (none of which are used in Figure 1).

AR: Text in lines 155-159 has been moved to the next "chapter" (The minimal responsive NLRP3 variant) so the text has a more clear flow or arguments.

10. Lines 264-267 are referring back to Fig 3 but have been inserted after the discussion of Figure 4 (lines

255-263).

AR: This text has been now joined to the previous chapter that it is referring to.

11. Supp Fig 14b and 14d are not referred to anywhere in the text.

AR: This has been corrected (those panels are now S14b and S14e due to insertion of new data).

12. Mini-NLRP3 (1-686 truncation) is referred to as mini-NLRP3 in the text but not in the figures. This is quite inconsistent for the reader.

AR: To make the manuscript easier to follow, MiniNLRP3 definition is now included in figure legends.

13. Line 1-2: Suggest: NLRP3 lacking THE leucine-rich repeat domain can be activated by canonical NLRP3 inflammasome stimuli.

AR: The authors are grateful to reviewer for suggestion; this grammatical error has been corrected, so we suggest as the new title: "NLRP3 lacking the leucine-rich repeat domain can be fully activated via the canonical inflammasome pathway".

We decided to keep the term "pathway" as the statement is also true for the disease-associated point mutants which are constitutively active.

14. Line 56: what does 'in certain ways' refer to here? The comparison of TLR4 agonists and NLRP3 stimuli does not make sense.

AR: Since our lab has a long track record on the research of TLR4, we have witnessed tens of the proposed direct TLR4 agonists, which from the biochemical point don't make sense and are now slowly being recognized as artefacts. In that sense the myriad of the proposed NLRP3 agonists is very similar, however, it is already being recognized that NLRP3 agonists affect some indirect processes (e.g. induction of K⁺ efflux, Munoz-Planillo et al., Immunity, 2013). However, as the manuscript is already a bit long, we have deleted this sentence so this is no longer relevant.

15. Line 94: 'apoptosome and inflammasome assembly' this needs to be referenced.

AR: The appropriate references were inserted.

16. Line 98: 'early reports' please reference these. 'the NLRP3 overexpression' delete 'the'.

AR: The grammatical issue has been resolved and references inserted.

17. Supp Fig 1: the legend is incorrect as for c and d it says: NLRPC4 (B) and NLRC2 (c)

AR: We thank the reviewer for noticing this error, which has been corrected.

18. Line 134 and Supp fig 3: There is clearly some variability in IL-6 and TNFa secretion with the various truncations (e.g. between 823 and 995 are lower than FL). These may be due to experimental variability, but unless statistics are performed on these data then it is inappropriate to say that 'none of the truncation variants affected the release levels'.

AR: What we intended to say is that the trend is similar for all variants and for the empty vector transduced cells. The release of those cytokines does not depend on NLRP3 either wild-type or variant.

We changed the sentence to: Introduction of NLRP3 either wild-type or truncation variant did not result in the substantial changes in the release of inflammasome-independent cytokine (IL-6 and TNF- α) (Supplementary Fig. 3).

19. Line 144: 'While THE protein level'.

AR: This has been corrected as suggested.

20. Line 145/Supp fig 5A: Where is the data showing removal of doxycycline results in decreased protein levels of 879 and 965? This blot would need to show levels prior to dox removal for comparison.

AR: The text was changed to: ... the protein levels of 1-879 and 1-965 were decreased upon the removal of doxycycline (as judged by the difference in the expression level at 3 and 6 h post doxycycline removal)

Here, we demonstrate that the original statement is correct, but we decided to retain the original figures, where all three variants are presented on the same blot.

21. Line 159: Supp fig 6b is called out before 6a.

AR: This has been corrected.

22. Line 173: 'the NLRC4' delete 'the'.

AR: This has been corrected as suggested.

23. Line 178: 'published studies' should be referenced.

AR: As we already exceeded the reference number limit for Nature Communications, we decided not to include those references as we find they are not essential for the manuscript and by exposing e.g. one reference would mean unjust "negative publicity", particularly since there are also studies which do not define the constructs they are working with at all.

The sentence has been changed to:

Variants containing predicted PYD-NBD-HD1-WHD were in the past quite commonly referred to as NLRP3 Δ LRR variants.

24. Figure 2e and 2f are too small and the x axis labels are not aligned – switch x axis labels to vertical.

AR: This has been changed as suggested.

25. Line 190 – same comment on Supp fig 7 as for Supp fig 3- there is some variation in TNF (Supp 7b) with the truncations.

AR: Text has been changed to:

Introduction of NLRP3 either as wild-type or truncation variant did not result in large changes in the release of inflammasome-independent cytokine...

26. Line 205 – the phenotype here is very similar but is not a ‘phenocopy’ or ‘the same phenotype’ Supp fig 8.

AR: Corrected as suggested.

27. Line 242 - ‘upon the NLRP3’ delete ‘the’.

AR: Corrected as suggested.

28. Figure 3e – what does %ASC speck/cell actually mean? Is this the percentage of the cells in the field of view that contain ASC specks? How many cells were analysed in total?

AR: ASC specks within the field are counted and divided by the number of nuclei (cells), and the explanation is provided in the figure legends. The number is practically the same as the number of ASC speck-containing cells (just a few cells contain more than a single speck). The number of cells analyzed in total: empty (198), empty+nig (240), 667(180), 667+nig(228), 688(213), 688+nig(192), FL(149), FL+nig(89). The number of cells is lower for nigericin-treated responsive variants as some dead cells have detached.

29. Figure 5c – the labelling here is not obvious – pulldowns with control or ATP beads need to be more clearly indicated.

AR: pull-downs are indicated and a legend is now included.

30. Line 308 – ‘of the NLRP3’ delete ‘the’. Also delete ‘similarly’.

AR: This has been corrected as suggested.

31. Supp Fig 12 d-f are called out before 12 c

AR: The panels in Suppl. Fig. 12 have been switched for this to be corrected.

32. Figure 6- Western blots (top panels) are not labelled.

AR: Corrected as suggested.

33. Line 333 ‘for the inflammasome’, delete ‘the’.

AR: Corrected as suggested.

34. Line 334 insert 'the' before 'shorter truncation' – 'THE shorter truncation'.

AR: Corrected as suggested.

35. Figure 7d – should be labelled as ASC-/- cells.

AR: Done as suggested.

36. Figure 8 a-d – what cells are these? Please label these figures it's impossible to tell what the difference is between a-d.

AR: Done as suggested.

37. Line 417 – 'the NLRP3', delete 'the'.

AR: Corrected as suggested.

38. Line 425 – insert the 'lacking THE LRR.'

AR: Corrected as suggested.

39. Line 457 – insert the 'THE present'

AR: Corrected as suggested.

REVIEWERS' COMMENTS:

Reviewer #3 (Remarks to the Author):

I congratulate the authors on this interesting manuscript. The manuscript revision addresses my concerns and in this Reviewer's opinion, is now ready for publication

REVIEWERS' COMMENTS: Reviewer #3 (Remarks to the Author):

I congratulate the authors on this interesting manuscript. The manuscript revision addresses my concerns and in this Reviewer's opinion, is now ready for publication Authors:

We would like to thank this reviewer for constructive remarks throughout the revision process.